# Understanding Virtual Nodes: Oversquashing and Node Heterogeneity

**Joshua Southern**[1,*,†], **Francesco Di Giovanni**[2,†],
**Michael Bronstein**[2,3], and **Johannes F. Lutzeyer**[4]

[1]*Imperial College London*
[2]*University of Oxford*
[3]*AITHYRA*
[4]*LIX, École Polytechnique, IP Paris*

## Abstract

While message passing neural networks (MPNNs) have convincing success in a range of applications, they exhibit limitations such as the oversquashing problem and their inability to capture long-range interactions. Augmenting MPNNs with a virtual node (VN) removes the locality constraint of the layer aggregation and has been found to improve performance on a range of benchmarks. We provide a comprehensive theoretical analysis of the role of VNs and benefits thereof, through the lenses of oversquashing and sensitivity analysis. First, we characterize, precisely, how the improvement afforded by VNs on the mixing abilities of the network and hence in mitigating oversquashing, depends on the underlying topology. We then highlight that, unlike Graph-Transformers (GTs), classical instantiations of the VN are often constrained to assign uniform importance to different nodes. Consequently, we propose a variant of VN with the same computational complexity, which can have different sensitivity to nodes based on the graph structure. We show that this is an extremely effective and computationally efficient baseline for graph-level tasks.

## 1 Introduction

Graph Neural Networks (GNNs) (Scarselli et al., 2009; Gori et al., 2005; Micheli, 2009) are a popular framework for learning on graphs. GNNs typically adopt a message passing paradigm (Gilmer et al., 2017), in which node features are aggregated over their local neighborhood recursively, resulting in architectures known as Message Passing Neural Networks (MPNNs). Whilst MPNNs have linear complexity in the number of edges and an often beneficial locality bias, they have been shown to have several limitations. In terms of expressivity, they are at most as powerful as the Weisfeiler-Leman test (Weisfeiler & Leman, 1968; Morris et al., 2019; Xu et al., 2018) and cannot count certain substructures (Chen et al., 2020; Bouritsas et al., 2023). Besides, the repeated aggregation of local messages can either make node representations indistinguishable—a process known as 'oversmoothing' (Nt & Maehara, 2019; Oono & Suzuki, 2020)—or limit the models' ability to capture long-range dependencies—a phenomenon called 'oversquashing' (Alon & Yahav, 2021).

In particular, oversquashing strongly depends on the graph-topology, since it describes the inability of MPNNs to reliably exchange information between pairs of nodes with large commute time (Di Giovanni et al., 2024). To overcome this limitation, methods have moved away from only performing aggregation over the local neighborhood and instead have altered the graph connectivity to reduce its commute time. Common examples include variations of multi-hop message-passing (Nikolentzos et al., 2020; Wang et al., 2020; Gutteridge et al., 2023), adding global descriptors (Horn et al., 2022), or rewiring operations based on spatial or spectral quantities (Topping et al., 2022; Arnaiz-Rodríguez et al., 2022). A notable case is given by Graph-Transformers (GTs) (Kreuzer et al., 2021; Ying et al., 2021), where global attention is used to weight the messages of each node pair.

---

[*]jks17@ic.ac.uk
[†]These authors contributed equally.

**Virtual Nodes**. While GTs incur a large memory cost, a significantly more efficient method that incorporates global information in each layer to combat oversquashing, is adding a virtual node (VN) connected to all nodes in the graph (Pham et al., 2017). This is widely used in practice (Gilmer et al., 2017; Battaglia et al., 2018) with empirical successes (Hu et al., 2020; Hwang et al., 2022). However, besides recent work connecting VNs to GTs (Cai et al., 2023; Rosenbluth et al., 2024), few efforts have been made to analyze in what way and to which extent VN improves the underlying model.

**Contributions**. We propose a theoretical study of VNs through the lenses of oversquashing and sensitivity analysis and highlight how they differ from approaches based on Graph-Transformers. Our analysis sheds light on the impact VNs have on reducing oversquashing by affecting the commute time, and the gap between VNs and GTs in terms of the sensitivity to the features of other nodes. Specifically, our main contributions are the following:

- In Section 3 we provide the first, systematic study of the impact of VNs on the oversquashing phenomenon. We show that, differently from more general rewiring techniques such as GTs, the improvement brought by VNs to the mixing abilities of the network, can be characterized and bounded in terms of the spectrum of the input graph.
- Relying on sensitivity analysis of node features, in Section 4 we find a gap between VNs and GTs in their ability to capture heterogeneous node importance. Our theory motivates a new formulation of MPNN + VN, denoted by MPNN + VN$_G$, that better leverages the graph structure to learn a heterogeneous measure of node relevance, at no additional cost.
- Finally, in Section 5, we first validate our theoretical analysis through extensive ablations and experiments. Next, we evaluate MPNN + VN$_G$ and show that it consistently surpasses the baseline MPNN + VN, precisely on those tasks where node heterogeneity matters.

We moreover want to remark that in Appendix B we make a further contribution, where we compare the expressivity of MPNN + VN and anti-smoothing techniques through polynomial filters. This allows us to contest prior belief that a reason for the success of MPNN + VN is their ability to simulate methods that prevent oversmoothing and to observe cases of 'beneficial smoothing' of node representations towards the final pooled representation. Since, this study could be considered to be of limited impact due to it being conducted in a linear setting, i.e., in absence of activation functions in the MPNN, we have chosen to not focus on this aspect of our work in this paper and only provide it in the appendix for especially interested readers.

## 2 PRELIMINARIES AND RELATED WORK

**Message Passing Neural Networks**. Let $G = (V, E)$ be an unweighted, undirected and connected graph with node set $V$ and edge set $E$. The connectivity of the graph is encoded in the adjacency matrix $\mathbf{A} \in \mathbb{R}^{n \times n}$, where $n = |V|$, and the 1-hop neighborhood of a node is $N_i = \{j \in V : (j, i) \in E\}$. Furthermore, node features $\{h_i : i \in V\} \subset \mathbb{R}^d$ are provided as input. Message passing layers are defined using learnable update and aggregation functions up and agg as follows

$$h_i^{(\ell+1)} = \mathsf{up}^{(\ell)}(h_i^{(\ell)}, \mathsf{agg}^{(\ell)}(\{h_j^{(\ell)} : j \in N_i\})), \tag{1}$$

where $h_i^{(\ell)}$ is the feature of node $i$ at layer $\ell$, up is a learnable update function (typically an MLP) and agg is invariant to node permutations. Below, we refer to models described as in (1) as MPNNs.

**Limitations of MPNNs**. Aggregating information over the 1-hop of each node iteratively, leads to systematic limitations. MPNNs without node identifiers (Loukas, 2019) are not universal (Maron et al., 2018) and are at most as powerful as the 1-WL test in distinguishing graphs (Morris et al., 2019; Xu et al., 2018). Additionally, when the task depends on features located at distant nodes (i.e. *long-range interactions*), MPNNs suffer from *oversquashing*, defined as the compression of information from exponentially growing receptive fields into fixed-size vectors (Alon & Yahav, 2021). In fact, oversquashing limits the MPNN's ability to solve tasks with strong interactions among nodes at large commute time (Di Giovanni et al., 2023a; Black et al., 2023; Di Giovanni et al., 2024). Finally, MPNNs may incur *oversmoothing*, whereby node features attain the same value as the number of layers increases (Nt & Maehara, 2019; Oono & Suzuki, 2020; Cai & Wang, 2020).

**Graph Transformers (GTs)**. An extension of the MPNN-class is given by Graph-Transformers (GTs) (Kreuzer et al., 2021; Rampášek et al., 2022; Ying et al., 2021; Hussain et al., 2022; Wu et al.,

2021; Chen et al., 2022), where a global attention mechanism connecting all pairs of nodes augments (or entirely replaces (Ma et al., 2023)) the 1-hop aggregation in MPNNs. GTs increase the expressive power of MPNNs by heavily relying on structural and positional encodings (Dwivedi et al., 2021), and entirely bypass the issue of oversquashing since the global attention breaks all potential bottlenecks arising from the graph topology. This comes at a price though, with GTs incurring quadratic memory cost and being sensitive to the choice of positional (structural) encodings. Furthermore, removing the locality constraint of MPNNs can cause GTs to lack a strong inductive bias which can lead to poor performance (Ma et al., 2023). To overcome this, Rampášek et al. (2022) combined a global attention mechanism with local message passing which led to improvements across multiple benchmarks. Their results suggest that maintaining the local inductive bias within message-passing is critical but that self-attention can improve performance further by capturing long-range interactions.

**MPNNs With Virtual Nodes**. An alternative approach to enhancing MPNNs without leading to the significant memory costs of GTs, consists in introducing a *virtual node* (VN) connected to all other nodes in the graph (Gilmer et al., 2017; Battaglia et al., 2018). A standard formulation of message passing with virtual nodes (Cai et al., 2023), referred to as MPNN + VN in the following, is:

$$
\begin{aligned}
h_{\mathsf{vn}}^{(\ell+1)} &= \mathsf{up}^{(\ell)}(h_{\mathsf{vn}}^{(\ell)}, \ \mathsf{agg}_{\mathsf{vn}}^{(\ell)}(\{h_j^{(\ell)} : j \in V\})), \\
h_i^{(\ell+1)} &= \mathsf{up}^{(\ell)}(h_i^{(\ell)}, \ \mathsf{agg}^{(\ell)}(\{h_j^{(\ell)} : j \in N_i\}), h_{\mathsf{vn}}^{(\ell)}).
\end{aligned}
\tag{2}
$$

The class MPNN + VN extends MPNNs in (1) to a multi-relational graph $G_{\mathsf{vn}}$ obtained by adding a VN connected to all $i \in V$, where any new edge is distinguished from the ones in the input graph.

**Understanding VN: Open Questions and Challenges**. Augmenting MPNNs with VNs has often been shown to improve performance (Hwang et al., 2022; Sestak et al., 2023). Nonetheless, a careful analysis of how and when VNs help is still lacking. Key related work was done by Cai et al. (2023), which relied on the universal approximation property of DeepSets (Segol & Lipman, 2020) to show that MPNN + VN with hidden state dimensions of $O(n^d)$ and $O(1)$ layers can approximate a self-attention layer. Their argument though discards the topology and holds in a non-uniform regime, raising questions about its practical implications. Crucially, VNs are mainly added to networks so as to model non-local interactions, as follows from (2) where any node pair is separated by at most 2 hops through the VN. Nonetheless, neither a formal analysis on the extent to which MPNN + VN mitigates oversquashing, nor a fine-grained comparison between GTs and MPNN + VN in terms of a sensitivity analysis (i.e. interactions among node features in a layer), have been conducted so far.

**Outline of This Work**. We provide a systematic study of MPNN + VN through the lenses of oversquashing and sensitivity analysis. We show that: The improvement brought by VN in mitigating oversquashing depends on the spectrum of the graph (Section 3); Differently from GTs, the sensitivity of VN to distinct node features is often uniform, and hence we propose $VN_G$, a new formulation of VN that can learn heterogeneous node importance at no additional cost (Section 4); The framework $VN_G$ consistently improves over VN, closing the empirical gap with GTs (Section 5).

**Graph-Level Tasks**. In this work, we focus on graph-level tasks; this way, we can compare the mixing abilities of MPNN + VN with that of MPNN, as per the graph-level analysis in Di Giovanni et al. (2024). Graph-level benchmarks (both classification and regression) offer a comprehensive test bed to compare VN and GTs, which is aligned with similar evaluations (Gutteridge et al., 2023; Barbero et al., 2024). Nonetheless, we emphasize that the statements in Section 4 hold layerwise, and hence do not depend on whether the task is graph-level or not; additionally, we can extend the results in Section 3 to node-level tasks following (Di Giovanni et al., 2024, Appendix E).

## 3 MPNN + VN and Oversquashing

VNs are often added to alleviate oversquashing, by reducing the diameter of the graph to 2. However, no formal analysis on the improvement brought by VNs to oversquashing has been derived thus far. In this section, we fill this gap in the literature and characterize how the spectrum of the input graph affects the impact of VN to commute time and hence the mixing abilities of the network. To study

oversquashing, we analyze general realizations of (2), whose layer updates have the form:

$$h_{\mathsf{vn}}^{(\ell+1)} = \sigma\Big(\boldsymbol{\Omega}_{\mathsf{vn}}^{(\ell)} h_{\mathsf{vn}}^{(\ell)} + \frac{1}{\tilde{n}} \sum_{j=1}^{n} \phi_{\mathsf{vn}}^{(\ell)}(h_{\mathsf{vn}}^{(\ell)}, h_j^{(\ell)})\Big),$$

$$h_i^{(\ell+1)} = \sigma\Big(\boldsymbol{\Omega}^{(\ell)} h_i^{(\ell)} + \sum_{j=1}^{n} \mathsf{A}_{ij}\psi^{(\ell)}(h_i^{(\ell)}, h_j^{(\ell)}) + \psi_{\mathsf{vn}}^{(\ell)}(h_i^{(\ell)}, h_{\mathsf{vn}}^{(\ell)})\Big), \tag{3}$$

where $\sigma$ is a pointwise nonlinear map, $\mathsf{A}$ is a (potentially) normalized adjacency matrix, $\boldsymbol{\Omega}$ is a weight matrix, $\psi, \psi_{\mathsf{vn}}, \phi_{\mathsf{vn}}$ are (learnable) message functions, and $\tilde{n}$ depends on our normalization choice (e.g. $\tilde{n} = 1$ for sum, or $\tilde{n} = n$ for mean). A key observation is that (3) coincide with the MPNNs analyzed in Di Giovanni et al. (2024), but operating on the multi-relational graph $G_{\mathsf{vn}}$ with adjacency

$$\mathbf{A}_{\mathsf{vn}} = \begin{pmatrix} \mathbf{A} & \mathbf{1} \\ \mathbf{1}^\top & 1 \end{pmatrix}, \quad \mathbf{1} = (1, \ldots, 1)^\top \in \mathbb{R}^n, \tag{4}$$

where edges connecting VN to nodes in $G$ have different type. Di Giovanni et al. (2023a); Black et al. (2023); Di Giovanni et al. (2024) have shown that oversquashing prevents the underlying model from exchanging information between nodes at large **commute time** $\tau$, where $\tau(i,j)$ measures the expected number of steps for a random walk to commute between $i$ and $j$. Accordingly, to *assess if and how a VN helps to mitigate oversquashing, we need to determine whether the commute time* $\tau_{\mathsf{vn}}$ *of* $G_{\mathsf{vn}}$ *is smaller than the commute time* $\tau$ *of the original graph* $G$. Below, we let $v_\ell$ be an orthonormal basis of eigenvectors for the graph Laplacian $\mathbf{L} = \mathbf{D} - \mathbf{A}$, with associated eigenvalues $0 = \lambda_0 < \lambda_1 \leq \ldots \leq \lambda_{n-1}$. The proof of the following Theorem and all subsequent results, can be found in Appendices E and F.

**Theorem 3.1.** *The commute time between nodes* $i, j$ *after adding a* VN *changes as*

$$\tau_{\mathsf{vn}}(i,j) - \tau(i,j) = 2|E| \sum_{\ell=1}^{n-1} \frac{1}{\lambda_\ell(\lambda_\ell + 1)}\Big(\frac{n}{|E|}\lambda_\ell - 1\Big)(v_\ell(i) - v_\ell(j))^2. \tag{5}$$

*In particular, the average change in commute time is:*

$$\frac{1}{n^2} \sum_{i,j=1}^{n}(\tau_{\mathsf{vn}}(i,j) - \tau(i,j)) = \frac{4|E|}{n} \sum_{\ell=1}^{n-1} \frac{1}{\lambda_\ell(\lambda_\ell + 1)}\Big(\frac{n}{|E|}\lambda_\ell - 1\Big). \tag{6}$$

The result in Theorem 3.1 highlights how the impact of adding a VN can be determined in terms of the spectrum of the input graph. While there are cases, e.g. when the graph is complete, where (6) is positive, for many real-world graphs adding a VN reduces the overall commute time: We empirically validate this claim in Section 5.1. Note that we prove Theorem 3.1 by calculating the explicit analytical form of the effective resistance between two nodes in $G_{\mathsf{vn}}$. We then exploit the fact that the commute time between nodes is proportional to their effective resistance. So, Theorem 3.1 can be trivially extended to make equivalent statements about the effective resistance between nodes instead of their commute time.

In our subsequent analysis we will make use of the notion of *mixing*, which was introduced by Di Giovanni et al. (2024). We begin by recalling its formal definition in Defintion 3.2.

**Definition 3.2** (Di Giovanni et al. (2024))**.** The quantity $\mathsf{mix}_y(i,j)$ is the mixing induced by a graph-level function among the features of nodes $i$ and $j$, and is defined as

$$\mathsf{mix}_y(i,j) = \max_{\mathbf{H}} \max_{1 \leq \alpha, \beta \leq d} \left| \frac{\partial^2 y(\mathbf{H})}{\partial h_i^\alpha \partial h_j^\beta} \right|.$$

Di Giovanni et al. (2024) made use of the mixing notion to formally assess the amount of interactions between $i$ and $j$ required by the underlying task, and compared it to the mixing induced by the MPNN-prediction after $m$ layers. In particular, they proved that oversquashing prevents MPNNs of bounded depth from solving tasks with required strong mixing between nodes at large commute time. Below, 'bounded weights' means that weight matrices and message functions derivatives have bounded spectral norms—see Appendix E for details.

**Theorem 3.3** (Adapted from Thm. 4.4 (Di Giovanni et al., 2024)). *There are graph-functions with mixing between nodes $i \neq j$ larger than some constant independent of $i, j$, such that for an* MPNN *of bounded weights to learn these functions, the number of layers $m$ must satisfy $m \geq \tau(i,j)/8$.*

For real-world graphs like peptides, small molecules, or images, (5) is negative (see Section 5.1). In light of Theorem 3.3, this means that adding a VN should reduce the minimal number of layers required to learn functions with strong mixing between nodes.

According to Di Giovanni et al. (2024), when the mixing induced by the MPNN after $m$ layers, denoted here as $\text{mix}^{(m)}(i,j)$, is smaller than the one required by the downstream task, then we have an instance of harmful oversquashing which prevents the network from learning the right interactions necessary to solve the problem at hand. We make this connection explicit in Corollary 3.4, in which we use Theorem 3.1 to characterize the improvement brought by a VN in terms of the spectrum of the Laplacian.

**Corollary 3.4.** *Given $G$ and nodes $i, j$ for which (5) is negative, there are graph-functions with mixing between $i \neq j$ larger than some constant independent of $i, j$, such that for an* MPNN + VN *of bounded weights to learn these functions, the minimal number of layers $m$ becomes*

$$m \geq \frac{\tau_{\mathsf{vn}}(i,j)}{8} - \frac{|E|}{8} \cdot \sum_{\ell=1}^{n-1} \frac{1}{\lambda_\ell(\lambda_\ell+1)} \Big( \frac{n}{|E|} \lambda_\ell - 1 \Big) (v_\ell(i) - v_\ell(j))^2. \tag{7}$$

The result in Corollary 3.4 shows that for real-world graphs where adding a VN significantly reduces the commute time—i.e., (5) is negative—the minimal number of layers required by MPNN + VN to learn graph functions with strong mixing among $i, j$, is smaller than that of MPNN. This is the first result showing the extent to which MPNN + VN alleviates oversquashing by increasing the mixing abilities of the network. Crucially, MPNN + VN cannot do better than Corollary 3.4, meaning that their efficacy is a function of the spectrum. While this implies that MPNN + VN may be sub-optimal, when compared to graph rewiring techniques such as GTs which can modify the commute time arbitrarily, the overall benefits of VN stem from their ability to reduce the commute time and hence mitigate oversquashing at very limited memory cost. We discuss next how we can further refine the formulation of VN to further close their gap with GTs, without increasing the computational complexity.

## 4 COMPARING MPNN + VN AND GTS THROUGH SENSITIVITY ANALYSIS

In Section 3 we showed that MPNN + VN can mitigate oversquashing and the extent to which this is possible. Another successful approach for combating oversquashing is given by Graph Transformers (GTs), that can entirely rewire the graph through the attention module. In fact, MPNN + VN are often used as a more efficient alternative to GTs (Cai et al., 2023). In this section we further compare MPNN + VN and GTs through sensitivity analysis and show that the single layer of a GT can learn a heterogeneous node scoring, while MPNN + VN generally cannot. In light of this analysis, we propose a simple variation of MPNN + VN, called MPNN + VN$_G$, which better uses the graph to learn a heterogeneous measure of node relevance while retaining the same efficiency as MPNN + VN. In our sensitivity analysis we will show MPNN + VN$_G$ to fall inbetween the fully homogeneous MPNN + VN and the potentially fully heterogeneous GTs.

**MPNN + VN Layers Are Homogeneous**. We start by reporting the layer update of the GPS architecture (Rampášek et al., 2022), one instance of the GT class that encodes both local and global information:

$$h_i^{(\ell+1)} = f^{(\ell)}(h_{i,\text{loc}}^{(\ell+1)} + \mathbf{Q}^{(\ell)} \sum_{k=1}^{n} a(h_i^{(\ell)}, h_k^{(\ell)}) h_k^{(\ell)}), \tag{8}$$

where $f$ is an MLP, $h_{\text{loc}}$ is the local update given by the choice of MPNN, and $a$ is the *attention* module. Depending on the available data and the chosen positional encoding scheme, GTs can capture heterogeneous relations. We can quantify such heterogeneity in a sensitivity analysis by deriving that the Jacobian $\partial h_i^{(\ell+1)}/\partial h_k^{(\ell)}$ is, in general, a function that depends on $k$, meaning that the state of node $i$ at layer $\ell+1$ is affected by the state of a node $k$ at the previous layer, as a function varying with $k$. Conversely, MPNN + VN depends on different nodes more uniformly. Explicitly, consider an MPNN + VN as in (3), where the VN update is

$$h_{\mathsf{vn}}^{(\ell+1)} = \sigma(\mathbf{\Omega}_{\mathsf{vn}}^{(\ell)} h_{\mathsf{vn}}^{(\ell)} + \mathbf{W}_{\mathsf{vn}}^{(\ell)} \mathsf{Mean}(\{h_j^{(\ell)}\})). \tag{9}$$

To exemplify the standard definition of VNs, we now provide the model equations of a GCN + VN.

$$h_{\mathsf{vn}}^{(\ell+1)} = \sigma(\mathbf{\Omega}_{\mathsf{vn}}^{(\ell)} h_{\mathsf{vn}}^{(\ell)} + \frac{1}{n}\sum_{j=1}^{n} \mathbf{W}_{\mathsf{vn}}^{(\ell)} h_j^{(\ell)}),$$

$$h_i^{(\ell+1)} = \sigma(\mathbf{\Omega}^{(\ell)} h_i^{(\ell)} + \sum_{j \in N_i} \frac{1}{\sqrt{d_i d_j}} \mathbf{W}^{(\ell)} h_j^{(\ell)} + h_{\mathsf{vn}}^{(\ell)}), \tag{10}$$

where $d_i$ denotes the node degree of node $i$ and $\mathbf{\Omega}_{\mathsf{vn}}^{(\ell)}, \mathbf{W}_{\mathsf{vn}}^{(\ell)}, \mathbf{W}^{(\ell)}$ denote trainable weight matrices. Given a node $k$ at a distance larger than 2 from node $i$, any message from node $k$ at layer $\ell - 1$ is first received by node $i$ at layer $\ell + 1$ through the VN.

**Proposition 4.1.** *For* MPNN + VN *whose* VN *update is* (9)*, the Jacobian* $\partial h_i^{(\ell+1)}/\partial h_k^{(\ell-1)}$ *is independent of $k$ whenever $k$ and $i$ are separated by more than 2 hops.*

We see that node $i$ receives a global yet homogeneous update from the VN, where the feature of each node $k$ at distance greater than 2 contribute the same to node $i$'s representation (after two layers). This in stark contrast to the case of GTs.

**MPNN + VN$_G$: A New Formulation Of VN**. Inspired by frameworks such as (8), allowing for more *heterogeneous* sensitivity through the VN, we propose MPNN + VN$_G$, a simple variation to MPNN + VN with the *same computational complexity*, $O(|E| + n)$, with layer updates of the form:

$$h_{i,\mathrm{loc}}^{(\ell+1)} = \mathsf{up}^{(\ell)}(h_i^{(\ell)}, \mathsf{agg}^{(\ell)}(\{h_j^{(\ell)} : j \in N_i\})), \tag{11}$$

$$h_{\mathsf{vn}}^{(\ell+1)} = \mathsf{up}_{\mathsf{vn}}^{(\ell)}(h_{\mathsf{vn}}^{(\ell)}, \mathsf{agg}_{\mathsf{vn}}^{(\ell)}(\{h_{j,\mathrm{loc}}^{(\ell+1)} : j \in V\})), \tag{12}$$

$$h_i^{(\ell+1)} = \widetilde{\mathsf{up}}^{(\ell)}(h_{i,\mathrm{loc}}^{(\ell+1)}, h_{\mathsf{vn}}^{(\ell+1)}). \tag{13}$$

Differently from MPNN + VN in (2), we first compute local updates based on the choice of aggregation (i.e. the underlying MPNN model) in (11), *then* we compute a global update through VN using such local representations in (12) (as illustrated in Appendix J), and finally we combine the local and non-local representations in (13). This asynchronous interleaving between a local and global update was previously shown to have a practical performance improvement over combining these updates in parallel (Rosenbluth et al., 2024; Yin & Zhong, 2023). To theoretically justify this improvement in our setting, we show how MPNN + VN$_G$ can assign more heterogeneous values to the global updates associated with the VN, by studying the sensitivity of node features. Similarly to (3) and (9), we consider an instance of MPNN + VN$_G$ of the form:

$$h_{i,\mathrm{loc}}^{(\ell+1)} = \sigma(\mathbf{\Omega}^{(\ell)} h_i^{(\ell)} + \sum_{j \in N_i} \psi_{ij}^{(\ell)}(h_i^{(\ell)}, h_j^{(\ell)})),$$

$$h_i^{(\ell+1)} = h_{i,\mathrm{loc}}^{(\ell+1)} + \mathsf{Mean}(\{\mathbf{Q}^{(\ell+1)} h_{j,\mathrm{loc}}^{(\ell+1)}\}). \tag{14}$$

To further exemplify our MPNN + VN$_G$ model, and in direct correspondance to (10), we provide the explicit model equations for the GCN+VN$_G$ model now. Please note that in our experiments in Section 5 we mostly work with the GatedGCN+VN and GatedGCN+VN$_G$ models. We provide the corresponding, slightly more complex, model equations in Appendix F.

$$h_{i,\mathrm{loc}}^{(\ell+1)} = \sigma(\mathbf{\Omega}^{(\ell)} h_i^{(\ell)} + \sum_{j \in N_i} \frac{1}{\sqrt{d_i d_j}} \mathbf{W}^{(\ell)} h_j^{(\ell)}),$$

$$h_i^{(\ell+1)} = h_{i,\mathrm{loc}}^{(\ell+1)} + \mathsf{Mean}(\{\mathbf{Q}^{(\ell+1)} h_{j,\mathrm{loc}}^{(\ell+1)}\}).$$

Differently from (2), two nodes now can exchange information after a single layer. Below, we let $z_i^{(\ell)}$ be the argument of $\sigma$ in (14), for each $i \in V$, and $\nabla_s$ be the Jacobian with respect to variable $s$.

**Proposition 4.2.** *Given $i \in V$ and $k \in V \setminus N_i$, the Jacobian $\partial h_i^{(\ell+1)}/\partial h_k^{(\ell)}$ computed using (14) is*

$$\frac{\partial h_i^{(\ell+1)}}{\partial h_k^{(\ell)}} = \frac{1}{n}\Big(\mathrm{diag}(\sigma'(z_k^{(\ell)}))(\mathbf{\Omega}^{(\ell)} + \sum_{u \in N_k} \nabla_1 \psi_{ku}^{(\ell)}(h_k^{(\ell)}, h_u^{(\ell)})) + \sum_{u \in N_k} \mathrm{diag}(\sigma'(z_u^{(\ell)}))\nabla_2\psi_{uk}^{(\ell)}(h_u^{(\ell)}, h_k^{(\ell)})\Big).$$

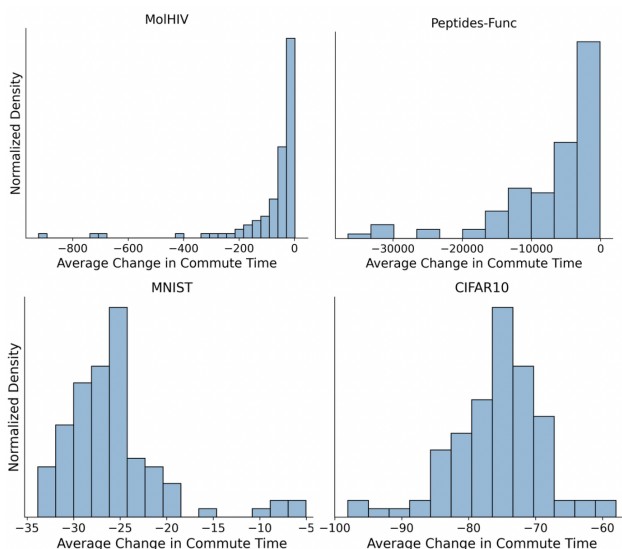

Figure 1: Effect of adding a virtual node on the average commute time of four graph datasets.

We see that the sensitivity of node $i$ to the feature of $k$ is a function of the graph-topology, i.e., the message functions evaluated over the neighbors of $k$. In particular, our proposed variation MPNN + VN$_G$ uses the same message functions $\psi_{ij}$ to learn a heterogeneous global update. Related to recent advances in graph pooling (Ranjan et al., 2020; Lee et al., 2019; Grattarola et al., 2022; Bianchi & Lachi, 2023), we utilize the graph structure in the global aggregator; nodes that are more relevant to each of their neighbors, are more likely to contribute more to the global update, instead of weighting each node the same as for MPNN + VN. We note that advantages of this approach depend on the choice of the underlying message function $\psi$ in (14) (see Table 8 in Appendix I).

It is interesting to note that the derivative in Proposition 4.2 does not depend on the index $i$ of the central node. If such a dependence was present, we would be working with a model that was essentially equivalent to a GT since the derivative of a given node $i$ depended both on $i$ and the currently considered neighbor $k$ (as shown in Proposition F.1 in Appendix F).

In light of our sensitivity analysis, we argue that for those tasks where an empirical gap between MPNN + VN and GT can be found, this is due to the latter leveraging its ability to assign heterogeneous node scores in its global update. Conversely, when no significant gap arises, we claim this is due to the task not requiring heterogeneous node scores. We validate these points in Section 5.1.

## 5 EXPERIMENTS

The purpose of this section is to first empirically verify the theoretical claims made in Sections 3 and 4. Then, we evaluate our proposed MPNN + VN$_G$ on a diverse set of benchmarks in Section 5.2.

### 5.1 ABLATIONS

Our ablations aim to validate the previous theoretical results, by answering the following questions:

**(Q1)** *How does adding a* VN *affect the average commute time on real-world graphs? [§3]*
**(Q2)** *Which tasks show a gap between* MPNN + VN *and GT due to node heterogeneity? [§4]*

To answer **(Q1)** we calculated (6) on 100 graphs from four commonly used real-world datasets covering peptides, small molecules and images. The distribution of the change of commute time across these four datasets is shown in Figure 1 and highlights the tendency for this change to be negative. This result demonstrates that adding a VN is highly beneficial on real-world graphs since it significantly reduces their commute time, thereby increasing their mixing abilities as per Corollary 3.4.

Table 1: Effects of projecting the non-local part of GPS onto the mean and its comparison to using a VN. Arrows indicate if the performance improves with higher (↑) or lower (↓) scores. We also report the standard deviation of the column sums in the first attention layer.

| Method | Pept-Func (↑) | Pept-Struct (↓) | MNIST (↑) | CIFAR10 (↑) |
|---|---|---|---|---|
| GPS | $0.6534 \pm_{.0091}$ | $0.2509 \pm_{.0014}$ | $98.051 \pm_{.126}$ | $\mathbf{72.298} \pm_{.356}$ |
| GPS + projection | $0.6498 \pm_{.0054}$ | $\mathbf{0.2487} \pm_{.0011}$ | $\mathbf{98.176} \pm_{.120}$ | $71.455 \pm_{.513}$ |
| GatedGCN+PE+VN | $\mathbf{0.6712} \pm_{.0066}$ | $\mathbf{0.2481} \pm_{.0015}$ | $98.122 \pm_{.102}$ | $70.280 \pm_{.380}$ |
| **std attention layer** | 0.0011 | 0.0007 | 0.0006 | 0.0038 |

We next aim to provide an empirical justification for the similar performances of MPNN+VN and GT (Cai et al., 2023) and answer **(Q2)**. To measure the heterogeneity of node relevance in the global update, we look at the similarity between the rows of the attention matrix in a GT. In fact, similar rows in $a(h_i^{(\ell)}, h_k^{(\ell)})$ in (8) correspond to assigning similar weights to each node representation, and hence less heterogeneity. To measure this, we calculate the mean standard deviation (std) of each column in $a(h_i^{(\ell)}, h_k^{(\ell)})$. We calculate this std on the first layer attention matrix, after training GPS with a GatedGCN as the base MPNN, on four real-world benchmarks from (Dwivedi et al., 2022) and (Dwivedi et al., 2023). In Table 1, we can see that for datasets where the std is small (Peptides-func, Peptides-struct, MNIST), using an MPNN + VN can match or even improve over the GPS implementation. On the other hand, CIFAR10 has a larger std, and we see that GPS outperforms the VN architecture. These results show that the performance gap between MPNN + VN and GTs is related to whether node heterogeneity is used by the GT on the benchmark, considering that MPNN + VN assigns equal sensitivity to different node features (Proposition 4.1). To further show the differences in the attention matrices and the amount of homogeneity on these tasks, we have visualized them in Appendix K. Additionally, we can measure the importance of heterogeneity for the task by removing the heterogeneity in the GT and seeing if we lose performance. We can do this in the GPS framework by setting all the rows of the attention matrix to be equal to their mean. We call this GPS + projection, and it forces the GPS model to become completely homogeneous. Again, we see that on datasets where the homogeneous GatedGCN+VN does well and the std of the attention matrix columns is low, we do not lose much performance with this mean projection. Whereas on CIFAR10, we find that the heterogeneity of nodes is important.

## 5.2 Evaluating Our Proposed MPNN + VN$_G$

We first evaluate MPNN + VN$_G$ on a diverse set of 5 graph-level datasets, comparing against classical MPNN benchmarks, Graph-Transformers (Ma et al., 2023; Shirzad et al., 2023; Rampášek et al., 2022; Kreuzer et al., 2021; Hussain et al., 2022), graph rewiring (Gutteridge et al., 2023), a random-walk based method (Tönshoff et al., 2023b), and a generalization of ViT/MLP-Mixer (He et al., 2023). Crucially, we also compare VN$_G$ with the standard implementation of VN (Cai et al., 2023).

**Experimental Details**. We evaluated MPNN + VN$_G$ on the **Long-Range Graph Benchmark** using a fixed 500k parameter budget and averaging over four runs. These *molecular* datasets (Peptides-Func, Peptides-Struct) have been proposed to test a method's *ability to capture long-range dependencies*. Additionally, we used two graph-level *image-based* datasets from **Benchmarking GNNs** (Dwivedi et al., 2023), where we run our model over 10 seeds. We also used a *code* dataset **MalNet-Tiny** (Freitas et al., 2020) consisting of function call graphs with up to 5,000 nodes. These graphs are considerably larger than previously considered graph-level benchmarks and showcase our *model's ability to improve over MPNN baselines on a large dataset*. We then evaluated our approach on three graph-level tasks from the **Open Graph Benchmark** (Hu et al., 2020), namely molhiv, molpcba and ppa. Details on datasets and training parameters used, can be found in Appendix G.

**Discussion**. We can see from Tables 2 and 3 that MPNN + VN$_G$ performs well across a variety of datasets and achieves the highest performance on Peptides-Struct, MNIST, ogbg-molhiv and ogbg-ppa. Furthermore, **MPNN + VN$_G$ improves over MPNN + VN on all tasks**, with the largest percentage improvement being shown on CIFAR10 (8.25%). The latter is perfectly aligned with the discussion in Sections 4 and 5.1, where we have described the importance of heterogeneity to the VN and observed that heterogeneity is particularly required on CIFAR10. Additionally, on the peptides datasets, which have small degrees but large diameters, we find that using MPNN + VN$_G$ can perform better than a

Table 2: Test performance on two LRGB datasets (Dwivedi et al., 2022) and three other benchmarks from (Dwivedi et al., 2023). For Peptides-Func and Peptides-Struct, ± std is shown over 4 runs whilst the remaining datasets are over 10 runs (missing values from literature are indicated by '-'). The first, second and third best results for each task are color-coded.

| Method | Pept-Func (↑) | Pept-Struct (↓) | MNIST (↑) | CIFAR10 (↑) | MalNet-Tiny (↑) |
|---|---|---|---|---|---|
| GCN | 0.5930 ±0.0023 | 0.3496 ±0.0013 | 90.705 ±0.218 | 55.710 ±0.381 | 81.0 |
| GINE | 0.5498 ±0.0079 | 0.3547 ±0.0045 | 96.485 ±0.252 | 55.255 ±1.527 | 88.98 ±0.56 |
| GatedGCN | 0.5864 ±0.0077 | 0.3420 ±0.0013 | 97.340 ±0.143 | 67.312 ±0.311 | 92.23 ±0.65 |
| GatedGCN+PE | 0.6765 ±0.0047 | 0.2477 ±0.0009 | - | 69.948 ±0.499 | - |
| GatedGCN+PE-ViT | 0.6942 ±0.0075 | 0.2465 ±0.0015 | 98.460 ±0.090 | 71.580 ±0.090 | - |
| GatedGCN+PE-Mixer | 0.6932 ±0.0017 | 0.2508 ±0.0007 | 98.320 ±0.040 | 70.600 ±0.220 | - |
| CRaWl | 0.7074 ±0.0032 | 0.2506 ±0.0022 | 97.940 ±0.050 | 69.010 ±0.259 | - |
| DRew | 0.7150 ±0.0044 | 0.2536 ±0.0015 | - | - | - |
| SAN+RWSE | 0.6439 ±0.0075 | 0.2545 ±0.0012 | - | - | - |
| EGT | - | - | 98.173 ±0.087 | 68.702 ±0.409 | - |
| GRIT | 0.6988 ±0.0082 | 0.2460 ±0.0012 | 98.108 ±0.111 | 76.468 ±0.881 | - |
| GPS | 0.6534 ±0.0091 | 0.2509 ±0.0014 | 98.051 ±0.126 | 72.298 ±0.356 | 93.50 ±0.41 |
| Exphormer | 0.6527 ±0.0043 | 0.2481 ±0.0007 | 98.414 ±0.035 | 74.690 ±0.125 | 94.02 ±0.21 |
| GatedGCN+PE+VN | 0.6712 ±0.0066 | 0.2481 ±0.0015 | 98.122 ±0.102 | 70.280 ±0.380 | 92.62 ±0.57 |
| GatedGCN+PE+VN$_G$ | 0.6822 ±0.0052 | 0.2458 ±0.0006 | 98.626 ±0.100 | 76.080 ±0.330 | 93.67 ±0.37 |

Table 3: Test performance on graph-level OGB benchmarks (Hu et al., 2020). Shown is the mean % ± std of 10 runs (missing values from literature are indicated with '-'). The first, second and third best results for each task are color-coded.

| Method | MolHIV (↑) | MolPCBA (↑) | PPA (↑) |
|---|---|---|---|
| GCN | 76.06 ±0.97 | 20.20 ±0.24 | 68.39 ±0.84 |
| GIN | 75.58 ±1.40 | 22.66 ±0.28 | 68.92 ±1.00 |
| DeeperGCN | 78.58 ±1.17 | 27.81 ±0.38 | 77.12 ±0.71 |
| CRaWL | - | 29.86 ±0.25 | - |
| SAN | 77.85 ±0.28 | 27.65 ±0.42 | - |
| GPS | 78.80 ±1.01 | 29.07 ±0.28 | 80.15 ±0.33 |
| GCN+VN | 75.99 ±1.19 | 24.24 ±0.34 | 68.57 ±0.61 |
| GIN+VN | 77.07 ±1.49 | 27.03 ±0.23 | 70.37 ±1.07 |
| GatedGCN+PE+VN | 76.87 ±1.36 | 28.48 ±0.26 | 80.27 ±0.26 |
| GatedGCN+PE+VN$_G$ | 79.10 ±0.86 | 29.17 ±0.27 | 81.17 ±0.30 |

variety of transformer-based approaches. Not only does this suggest that MPNN + VN$_G$ works well in practice but that complex long-range dependencies might not play a primary role for these tasks. This was also supported by Tönshoff et al. (2023a) where the lower performance of MPNNs compared to GTs was found to be caused by insufficient hyperparameter tuning. We also show strong performance on the larger MalNet-Tiny dataset. Previous work has shown that using a full transformer outperforms linear attention-based methods (Rampášek et al., 2022) on this task. However, we show that we can match the performance of the full transformer and still be more computationally efficient.

## 6 CONCLUSION

In Section 3, we characterize how the spectrum of the input graph affects the impact of VN on the commute time and hence to oversquashing. We have also evaluated this on real-world benchmarks in Section 5.1. In Section 4, we show that standard VN implementations cannot assign different scores to nodes in the global update. Consequently, we propose MPNN + VN$_G$, a variation of MPNN + VN sharing the same computational complexity, that uses the graph to learn heterogeneous node importance. In Section 5, we show that this outperforms MPNN + VN on a range of benchmarks, precisely corresponding to those where GTs outperform MPNN + VN by learning heterogenos node scores.

**Limitations and Future Work.** We have not empirically explored the change in commute time afforded by GT and rewiring-based approaches. Future work can compare these approaches and VN, to assess their mixing abilities. We furthermore relegate the comparison of different message passing mechanisms in the $\text{VN}_G$ to future work since this would unduly extend the scope of our work.

ACKNOWLEDGMENTS

J.S. is supported by the UKRI CDT in AI for Healthcare http://ai4health.io (EP/S023283/1). M.B. is supported by EPSRC Turing AI World-Leading Research Fellowship No. EP/X040062/1 and EPSRC AI Hub on Mathematical Foundations of Intelligence: An "Erlangen Programme" for AI No. EP/Y028872/1. J.L. is supported by the French National Research Agency (ANR) via the "GraspGNNs" JCJC grant (ANR-24-CE23-3888).

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

## A  OUTLINE OF THE APPENDIX

In Appendix B we study VNs in the context of the oversmoothing problem and for better readibility we provide the proofs of the theoretical statements in Appendix C. Then Appendix D contains further theoretical results connecting VN and the choice of global pooling. Next, we report the analysis on commute time and oversquashing when adding a VN in Appendix E. Proofs for the sensitivity analysis results in Section 4 are finally reported in Appendix F.

Regarding additional ablations and experimental details, we comment on training details, hyperparameters, datasets, and hardware in Appendix G. Next, we describe additional experiments confirming that VN introduces smoothing on graph-level tasks in Appendix H, focusing on the impact of the final pooling. We further validate the role played by the underlying MPNN model in MPNN + $VN_G$ in Appendix I, propose a visualization of the framework in Appendix J, and finally report attention maps learned by Graph-Transformer on benchmarks—used as a proxy-measure for the heterogeneity required by the task—in Appendix K.

## B  MPNN + VN AND SMOOTHING

Despite the empirical success of MPNN + VN, the reasons behind these improvements still lack a thorough understanding. To this aim, Cai et al. (2023) postulated that MPNN + VN is also beneficial due to its ability to replicate PairNorm (Zhao & Akoglu, 2020), a framework designed to mitigate oversmoothing. In this appendix we provide theoretical arguments against this conjecture, proving that MPNN + VN loses expressive power when it emulates PairNorm.

**Comparison to PairNorm**. Methods such as PairNorm (Zhao & Akoglu, 2020), aim at mitigating *oversmoothing*, which occurs when the node representations converge to the same vector in the limit of many layers, independent of the task (Nt & Maehara, 2019; Oono & Suzuki, 2020). To address the claims in Cai et al. (2023) and compare MPNN + VN and PairNorm, in this section we adopt the assumptions in Zhao & Akoglu (2020) and other theoretical works on oversmoothing (Di Giovanni et al., 2023b), and consider networks whose layers are linear. In particular, to assess advantages of MPNN + VN over PairNorm, we study their expressivity. Since aggregating $m$ linear layers yields a polynomial filter of the form $p(\mathbf{A})\mathbf{H}$, where $\mathbf{H}$ are the input node features, we choose as expressivity metric precisely the ability of a model to learn such polynomial filters. In fact, under mild assumptions, polynomial filters are comparable to the 1-WL test (Wang & Zhang, 2022). Additionally, when the input features $\mathbf{H}$ are constant, polynomial filters (17) coincide with **graph moments**, a metric of expressivity studied in Dehmamy et al. (2019), further validating our approach.

**Measuring Expressivity With Polynomial Filters**. Since in this section we restrict the analysis to models that have linear layers, we introduce a simple MPNN baseline, which is obtained from (1) by choosing a sum aggregation and a linear update with a residual connection:

$$\mathbf{H}^{(\ell+1)} = \mathbf{H}^{(\ell)} + \mathbf{A}\mathbf{H}^{(\ell)}\mathbf{W}^{(\ell)}, \tag{15}$$

where $\mathbf{W}^{(\ell)}$ is a $d \times d$ weight matrix and $\mathbf{H}^{(\ell)}$ is the $n \times d$ feature matrix at layer $\ell$. Note that the choice of the sum aggregation in the MPNN, i.e., the occurence of $\mathbf{A}$ in (15) and the subsequent analysis, is made without loss of generality and our analysis applies to any other message passing operator used instead of $\mathbf{A}$. Similarly, for MPNN + VN, we consider a layer update with mean aggregation for VN, akin to Cai et al. (2023):

$$\mathbf{H}^{(\ell+1)} = \mathbf{H}^{(\ell)} + \mathbf{A}\mathbf{H}^{(\ell)}\mathbf{W}^{(\ell)} + \frac{1}{n}\mathbf{1}\mathbf{1}^\top\mathbf{H}^{(\ell-1)}\mathbf{Q}^{(\ell)}, \tag{16}$$

with $\mathbf{Q}^{(\ell)}$ a weight matrix and $\mathbf{1} \in \mathbb{R}^n$ the vector of ones. For simplicity, we assume that the output after $m$ layers $\mathbf{Y}_m$ also has dimension $d$ and choose a *mean* pooling operation in the final layer, so that given a weight matrix $\mathbf{\Theta}$ we have $\mathbf{Y}_m = \mathbf{\Theta}\mathsf{Mean}(\mathbf{H}^{(m)})$, with $\mathbf{H}^{(m)}$ computed by either (15) or (16). We can express $\mathbf{Y}_m$ as a graph-level polynomial filter in terms of $d \times d$ weight matrices $\{\mathbf{\Theta}_k\}$:

$$\mathbf{Y}_m = \mathsf{Mean}\big(\sum_{k=0}^{m} \mathbf{A}^k\mathbf{H}\mathbf{\Theta}_k\big)^\top. \tag{17}$$

We note now that MPNN + VN (16) recovers PairNorm when $\mathbf{Q}^{(\ell)} = -\mathbf{I}$, with $\mathbf{I}$ the $d \times d$-identity matrix. We denote this special choice as MPNN - VN.

**Theorem B.1.** *There are polynomial filters (17) that can be learned by* MPNN *but not by* MPNN - VN. *Conversely, any polynomial filter learned by* MPNN *(15) can also be learned by* MPNN + VN *(16). Furthermore, there exist polynomial filters that can be learned by* MPNN + VN *but not by* MPNN.

Theorem B.1 implies that the ability of an MPNN - VN to replicate PairNorm comes at the cost of expressivity. Conversely, MPNN + VN (when they do not replicate PairNorm) are more expressive than MPNN when learning polynomial filters. In light of Theorem B.1, we argue that, contrary to the claim in Cai et al. (2023), MPNN + VN generally avoids replicating PairNorm to maintain its greater expressivity. We empirically validate this statement in Appendix H.1 and report the smoothing effects of VN on common benchmarks in Appendix H.2.

In summary, our analysis suggests that on graph-level tasks, benefits of MPNN + VN are independent of their ability to replicate methods such as PairNorm. In fact, for a *graph-level* prediction, whose output is computed using a global mean, some 'smoothing' is actually unavoidable. In particular, the *mean* pooling in (17) corresponds to projecting the final features precisely onto the subspace spanned by $\mathbf{1} \in \mathbb{R}^n$, i.e., the kernel of the (unnormalized) graph Laplacian associated with oversmoothing (Cai & Wang, 2020; Di Giovanni et al., 2023b). This allows us to clarify a key difference when studying oversmoothing for node-level and graph-level tasks. In the former case, if two node features become indistinguishable, then they will be assigned the same label, which is often independent of the task, thereby resulting in **harmful oversmoothing**. Conversely, in the latter case, even if the node features converge to the same representation, this behavior may result in **beneficial oversmoothing** if such common representation is controlled to match the desired, *global* output. We argue that for this reason, oversmoothing has hardly been observed as an issue on graph-level tasks. We refer to Proposition D.1 in Appendix C for results relating VN to the choice of pooling and to ablations in Appendix H.2.

## C PROOFS OF RESULTS IN APPENDIX B

*Proof of Theorem B.1.* First, we note that similarly to Cai et al. (2023), the VN state is initialized to be zero, i.e., the first layer of Equation (16) is

$$\mathbf{H}^{(1)} = \mathbf{H} + \mathbf{A}\mathbf{H}\mathbf{W}^{(0)} \tag{18}$$

where $\mathbf{H}$ is matrix of input node features. It is trivial to check that any polynomial filter that can be learned by an MPNN can also be learned by MPNN + VN, considering that if we set the additional weight matrices $\{\mathbf{Q}_\ell\}$ to be zero in (16), then we recover the MPNN case in (15). Accordingly, to prove that MPNN + VN is strictly more expressive than MPNN when learning polynomial filters, we only need to show that there exist $m \in \mathbb{N}$ and polynomial filters of degree $m$ that can be learned by an MPNN + VN of $m$ layers, but not by MPNN. We show that such a polynomial can already be found when $m = 2$. Namely, given a linear 2-layer MPNN + VN, we can write the 2-layer representation $\mathbf{H}^{(2)}$ explicitly as

$$\mathbf{H}^{(2)} = \mathbf{H} + \mathbf{A}\mathbf{H}(\mathbf{W}^{(0)} + \mathbf{W}^{(1)}) + \mathbf{A}^2\mathbf{H}\mathbf{W}^{(0)}\mathbf{W}^{(1)} + \frac{1}{n}\mathbf{1}\mathbf{1}^\top\mathbf{H}\mathbf{Q}^{(1)}. \tag{19}$$

Recall that given the choice of global mean pooling, the final output is

$$\mathbf{Y}_2 = \tilde{\mathbf{\Theta}}\mathsf{Mean}(\mathbf{H}^{(2)}). \tag{20}$$

Since $\mathsf{Mean}(\mathbf{H}^{(2)}) = (\mathbf{H}^{(2)})^\top\mathbf{1}/n$, we find

$$\mathbf{Y}_2 = \mathbf{\Theta}_0\mathsf{Mean}(\mathbf{H}) + \mathbf{\Theta}_1\mathsf{Mean}(\mathbf{A}\mathbf{H}) + \mathbf{\Theta}_2\mathsf{Mean}(\mathbf{A}^2\mathbf{H}),$$

where the weights characterizing the family of quadratic polynomial filters are defined by the system:

$$\mathbf{\Theta}_0 = \tilde{\mathbf{\Theta}}\Big(\mathbf{I} + (\mathbf{Q}^{(1)})^\top\Big), \tag{21}$$

$$\mathbf{\Theta}_1 = \tilde{\mathbf{\Theta}}\Big((\mathbf{W}^{(0)})^\top + (\mathbf{W}^{(1)})^\top\Big), \tag{22}$$

$$\mathbf{\Theta}_2 = \tilde{\mathbf{\Theta}}\Big((\mathbf{W}^{(1)})^\top(\mathbf{W}^{(0)})^\top\Big). \tag{23}$$

Assume now that the class of polynomial filters we wish to learn, have a vanishing zeroth-order term, i.e., $\boldsymbol{\Theta}_0 = \mathbf{0}$. **For the MPNN not containing a VN**, corresponding to $\mathbf{Q}^{(1)} = \mathbf{0}$ in (21), then we must have $\tilde{\boldsymbol{\Theta}} = \mathbf{0}$. Therefore, the only quadratic polynomial without a zeroth-order term that can be learned by a linear MPNN as in (15) of degree 2 is the trivial polynomial. Conversely, an MPNN + VN can, for example, learn the weights $\mathbf{Q}^{(1)} = -\mathbf{I}$ to ensure $\boldsymbol{\Theta}_0 = \mathbf{0}$, and have non-trivial first and second-order terms using the $\mathbf{W}$-weight matrices. *This proves that on any given graph there exist polynomial filters that can be learned by* MPNN + VN *but not by* MPNN.

To conclude the proof, we now need to show that MPNN + VN loses expressive power when it emulates PairNorm. We observe that for the case of MPNN - VN, i.e., $\mathbf{Q}^{(\ell)} = -\mathbf{I}$ for $\ell \in \mathbb{N}$, after two layers, the term $\boldsymbol{\Theta}_0$ is always zero. Since MPNN after two layers can learn quadratic polynomial filters with non-trivial zero-th order terms, this completes the proof. $\qquad\square$

We want to remark that MPNN-VN is a slight simplification of PairNorm since it firstly, does not include the scaling factor that is present in the originally proposed PairNorm method and secondly, does not use the updated hidden representations in the substracted mean at every later. However, the PairNorm scaling factor can be absorbed into $\tilde{\boldsymbol{\Theta}}$ in (20). We furthermore observe that when using the original PairNorm normalisation, then every layer (by design) has zero mean, which in particular implies that no graph moment of order $m$ can be learned after $m$ layers, i.e., the $m$-th coefficient always vanishes. Hence, our statements also extend to the originally proposed PairNorm.

Note further that Theorem B.1 can be extended to apply to MPNNs with an linear embedding layer, i.e., MPNNs where (15) becomes

$$\mathbf{H}^{(1)} = \mathbf{H}\mathbf{W} + \mathbf{A}\mathbf{H}^{(1)}\mathbf{W}^{(1)},$$
$$\mathbf{H}^{(\ell+1)} = \mathbf{H}^{(\ell)} + \mathbf{A}\mathbf{H}^{(\ell)}\mathbf{W}^{(\ell)} \qquad \text{for } \ell > 1. \tag{24}$$

**Theorem C.1.** *There are polynomial filters (17) that can be learned by* MPNN *with embedding layers as defined in (24) but not by* MPNN - VN. *Conversely, any polynomial filter learned by* MPNN *(24) can also be learned by* MPNN + VN *with an analogous embedding layer on the initial node features. Furthermore, there exist polynomial filters that can be learned by* MPNN + VN *but not by* MPNN *when both models have a linear embedding layer on the initial node features.*

*Proof.* The structure of this proof follows the Proof of Theorem B.1.

Again, it is trivial to check that any polynomial filter that can be learned by an MPNN can also be learned by MPNN + VN, considering that if we set the additional weight matrices $\{\mathbf{Q}_\ell\}$ to be zero, then we recover the MPNN case in (24).

If we consider the MPNN in (24) then the system of equations in (21) becomes

$$\boldsymbol{\Theta}_0 = \tilde{\boldsymbol{\Theta}}\Big((\mathbf{W})^T + (\mathbf{Q}^{(1)})^T(\mathbf{W})^T\Big),$$
$$\boldsymbol{\Theta}_1 = \tilde{\boldsymbol{\Theta}}\Big((\mathbf{W}^{(0)})^T(\mathbf{W})^T + (\mathbf{W}^{(1)})^T(\mathbf{W})^T\Big),$$
$$\boldsymbol{\Theta}_2 = \tilde{\boldsymbol{\Theta}}\Big((\mathbf{W}^{(1)})^T(\mathbf{W}^{(0)})^T(\mathbf{W})^T\Big).$$

If we now consider polynomial filters with a vanishing first-order term and an invertible second-order term, we obtain $(\mathbf{W}^{(0)})^T = -(\mathbf{W}^{(1)})^T$. Consequently, for an MPNN without VN, we have

$$(\boldsymbol{\Theta}_0)^{-1}\boldsymbol{\Theta}_2 = -((\mathbf{W})^T)^{-1}(\mathbf{W}^{(1)})^T(\mathbf{W}^{(1)})^T(\mathbf{W})^T.$$

Hence,

$$\det((\boldsymbol{\Theta}_0)^{-1}\boldsymbol{\Theta}_2) = (-1)^d\det(\mathbf{W}^{(1)})^2. \tag{25}$$

Therefore, the polynomial filters an MPNN can learn are constrained, since the sign of the determinant in (25) is fully determined by the hyperparameter $d$. Without loss of generality, for an even $d$ an MPNN cannot learn the polynomial $\hat{\Theta}_0 = \hat{\Theta}_2 = I$ and $\hat{\Theta}_1 = 0$. On the other hand, this polynomial can be learned by MPNN+VN by learning $\tilde{\boldsymbol{\Theta}} = I$, $(\mathbf{W}^{(0)})^T = -(\mathbf{W}^{(1)})^T$, $(\mathbf{W})^T = (-(\mathbf{W}^{(1)})^T(\mathbf{W}^{(1)})^T)^{-1}$ and $(\mathbf{Q}^{(1)})^T = (\mathbf{I} - \mathbf{W})^T)\mathbf{W})^T$. $\qquad\square$

## D    ADDITIONAL RESULTS ON THE ROLE OF POOLING

In this section we discuss how the choice of the global pooling affects the benefits of VN. Consider an alternative global pooling, which instead of projecting the final features onto $\mathbf{1}/n$ (i.e. computing the mean), projects the final-layer features onto $\mathbf{v} \in \mathbb{R}^n$ satisfying $\mathbf{v}^\top \mathbf{1} = 0$,

$$\mathbf{Y}_m^{\mathbf{v}} = \tilde{\mathbf{\Theta}}(\mathbf{H}^{(m)})^\top \mathbf{v} = \sum_{k=0}^{m} \mathbf{\Theta}_k (\mathbf{A}^k \mathbf{H})^\top \mathbf{v}. \tag{26}$$

We can then show that there exist polynomial filters that MPNN + VN is no longer able to express when we replace mean pooling with $\mathbf{v}$-pooling.

**Proposition D.1.** *Given $\mathbf{v}$ orthogonal to $\mathbf{1}$, a linear MPNN + VN of $m$ layers can generate weights $\{\mathbf{\Theta}_k\}$ in the output $\mathbf{Y}_m$ in (17) that cannot be generated by a linear MPNN + VN using a global pooling induced by $\mathbf{v}$, as in (26).*

*Proof of Proposition D.1.* Once again, we show that it suffices to consider the case of quadratic polynomial filters, i.e., $m = 2$ given in the system of equations in (21). However, since $\mathbf{1}$ and $\mathbf{v}$ are orthogonal, then there is no $\mathbf{Q}^{(1)}$-term in Equation (21). As such, we see that the quadratic polynomial $\mathbf{Y}_2^{\mathbf{v}}$ has zero-th order coefficient vanishing if and only if $\tilde{\mathbf{\Theta}} = \mathbf{0}$, i.e., the quadratic polynomial is trivial. Conversely, MPNN + VN can instead learn non-trivial quadratic polynomial without zero-th order term by simply setting $\mathbf{Q}^{(1)} = -\mathbf{I}$. $\qquad\square$

**A Fourier Perspective on Pooling.** Proposition D.1 highlights that MPNN + VN may lose the ability to express polynomial filters when we no longer have alignment between VN and the global pooling. This result raises a *an important question*: What is the bias resulting from a certain choice of global pooling? Consider linear MPNNs as above where the global pooling is obtained by projecting onto some $\mathbf{v} \in \mathbb{R}^n$, with the case $\mathbf{v} = \mathbf{1}$ recovering the mean (sum) pooling. Given $\{\psi_i\} \subset \mathbb{R}^n$, a basis of orthonormal eigenvectors of $\mathbf{A}$, with eigenvalues $\{\lambda_i\}$, we write $\mathbf{v} = \sum_i c_i \psi_i$, where $\{c_i\}$ are the graph Fourier coefficients of $\mathbf{v}$. Given $1 \leq r \leq d$, the channel $r$ of the final output as per (26) takes on the following form in the Fourier domain

$$y_r = \sum_{k=0}^{m} \sum_{p=1}^{d} (\Theta_k)_{rp} \sum_{i=0}^{n-1} c_i \lambda_i^k \langle h^p, \psi_i \rangle, \tag{27}$$

where $h^p \in \mathbb{R}^n$ is the channel $p$ of the input features. Accordingly, the Fourier coefficients $\{c_i\}$ of $\mathbf{v}$ and the associated frequencies $\{\lambda_i\}$, determine which projection $\langle h^p, \psi_i \rangle$ affects the output the most. The fact that on many datasets mean (sum) pooling is beneficial, and hence adding VN is mostly advantageous as per Proposition D.1, highlights that the underlying tasks depend on the projection of the features onto those eigenvectors $\{\psi_i\}$ aligned with the vector $\mathbf{1}$.

## E    PROOFS OF RESULTS IN SECTION 3

In this section, we report the proofs of the theoretical results in Section 3 along with additional details. We start by proving Theorem 3.1.

*Proof of Theorem 3.1.* We start by recalling that the effective resistance between two nodes $i$ and $j$, denoted by $R(i, j)$, can be computed as (Ghosh et al., 2008):

$$R(i, j) = L_{ii}^\dagger + L_{jj}^\dagger - 2L_{ij}^\dagger, \tag{28}$$

where $\mathbf{L}^\dagger$ is the pseudo-inverse of the graph Laplacian, i.e.,

$$\mathbf{L}^\dagger = \sum_{\ell=1}^{n-1} \frac{1}{\lambda_\ell} v_\ell v_\ell^\top. \tag{29}$$

Accordingly, we can write the effective resistance between two nodes $i$ and $j$ explicitly as

$$R(i, j) = \sum_{\ell=1}^{n-1} \frac{1}{\lambda_\ell} (v_\ell(i) - v_\ell(j))^2. \tag{30}$$

Naturally, for the graph $G_{\text{vn}}$ we obtain from adding VN, the formula changes as

$$R_{\text{vn}}(i,j) = \sum_{\ell=1}^{n} \frac{1}{\lambda_\ell^{\text{vn}}} (v_\ell^{\text{vn}}(i) - v_\ell^{\text{vn}}(j))^2, \tag{31}$$

where $\mathbf{L}_{\text{vn}} v_\ell^{\text{vn}} = \lambda_\ell^{\text{vn}} v_\ell^{\text{vn}}$. When we add the virtual node though, the adjacency matrix changes as in Equation (4), meaning that the graph Laplacian can be written as

$$\mathbf{L}_{\text{vn}} = \mathbf{D}_{\text{vn}} - \mathbf{A}_{\text{vn}} = \begin{pmatrix} \mathbf{D} + \mathbf{I} & \mathbf{0} \\ \mathbf{0}^\top & n+1 \end{pmatrix} - \begin{pmatrix} \mathbf{A} & \mathbf{1} \\ \mathbf{1}^\top & 1 \end{pmatrix} = \begin{pmatrix} \mathbf{L} + \mathbf{I} & -\mathbf{1} \\ -\mathbf{1}^\top & n \end{pmatrix}. \tag{32}$$

We recall now that $v_0 = \frac{1}{\sqrt{n}}\mathbf{1}$, i.e., $\mathbf{L1} = \mathbf{0}$. Accordingly, we find that $v_0^{\text{vn}} = \frac{1}{\sqrt{n+1}} \begin{pmatrix} \mathbf{1} \\ 1 \end{pmatrix}$, i.e., $\mathbf{L}_{\text{vn}} v_0^{\text{vn}} = \mathbf{0}$. Similarly, from the block decomposition of $\mathbf{L}_{\text{vn}}$ we derive that

$$\mathbf{L}_{\text{vn}} \begin{pmatrix} v_\ell \\ 0 \end{pmatrix} = \begin{pmatrix} (\mathbf{L}+\mathbf{I})v_\ell \\ -\mathbf{1}^\top v_\ell \end{pmatrix} = (\lambda_\ell + 1) \begin{pmatrix} v_\ell \\ 0 \end{pmatrix}, \tag{33}$$

where we have used that $v_\ell$ are orthogonal to $v_0 = \frac{1}{\sqrt{n}}\mathbf{1}$ for all $\ell \geq 1$. Accordingly $\begin{pmatrix} v_\ell \\ 0 \end{pmatrix}$ are eigenvectors of $\mathbf{L}_{\text{vn}}$ for all $1 \leq \ell \leq n-1$ with eigenvalue $\lambda_\ell + 1$. Finally we note that

$$\mathbf{L}_{\text{vn}} \frac{1}{\sqrt{n^2+n}} \begin{pmatrix} \mathbf{1} \\ -n \end{pmatrix} = \frac{1}{\sqrt{n^2+n}} \begin{pmatrix} (\mathbf{L}+\mathbf{I})\mathbf{1} + n\mathbf{1} \\ -\mathbf{1}^\top\mathbf{1} - n^2 \end{pmatrix} = (1+n) \frac{1}{\sqrt{n^2+n}} \begin{pmatrix} \mathbf{1} \\ -n \end{pmatrix}, \tag{34}$$

which means that $\frac{1}{\sqrt{n^2+n}} \begin{pmatrix} \mathbf{1} \\ -n \end{pmatrix}$ is the final eigenvector of $\mathbf{L}_{\text{vn}}$ with eigenvalue $n+1$. We can then use the spectral decomposition of $\mathbf{L}_{\text{vn}}$ to write the effective resistance between nodes $i$ and $j$ after adding VN as

$$R_{\text{vn}}(i,j) = \sum_{\ell=1}^{n} \frac{1}{\lambda_\ell^{\text{vn}}} (v_\ell^{\text{vn}}(i) - v_\ell^{\text{vn}}(j))^2 = \sum_{\ell=1}^{n-1} \frac{1}{\lambda_\ell + 1} (v_\ell(i) - v_\ell(j))^2 + \frac{1}{n+1} \frac{1}{n^2+n} (1-1)^2$$

$$= \sum_{\ell=1}^{n-1} \frac{1}{\lambda_\ell + 1} (v_\ell(i) - v_\ell(j))^2.$$

Finally, we recall that on a graph, the commute time $\tau$ is proportional to the effective resistance (Lovász, 1993), i.e.,

$$\tau(i,j) = 2|E|R(i,j).$$

Accordingly, we can use that on $G_{\text{vn}}$ the number of edges is equal to $|E| + n$ and compare the commute time after adding VN with the baseline case:

$$\tau_{\text{vn}}(i,j) - \tau(i,j) = \sum_{\ell=1}^{n-1} \left( \frac{2(|E|+n)}{\lambda_\ell + 1} - \frac{2|E|}{\lambda_\ell} \right) (v_\ell(i) - v_\ell(j))^2$$

$$= 2|E| \sum_{\ell=1}^{n-1} \frac{1}{\lambda_\ell(\lambda_\ell + 1)} \left( \frac{n}{|E|}\lambda_\ell - 1 \right) (v_\ell(i) - v_\ell(j))^2,$$

which is precisely Equation (5). We can the sum over all pairs of nodes to obtain

$$\sum_{i,j=1}^{n} (\tau_{\text{vn}}(i,j) - \tau(i,j)) = 2|E| \sum_{i,j=1}^{n} \sum_{\ell=1}^{n-1} \frac{1}{\lambda_\ell(\lambda_\ell + 1)} \left( \frac{n}{|E|}\lambda_\ell - 1 \right) (v_\ell(i) - v_\ell(j))^2$$

$$= 2|E| \sum_{\ell=1}^{n-1} \frac{1}{\lambda_\ell(\lambda_\ell + 1)} \left( \frac{n}{|E|}\lambda_\ell - 1 \right) \sum_{i,j=1}^{n} (v_\ell(i) - v_\ell(j))^2$$

$$= 2|E| \sum_{\ell=1}^{n-1} \frac{1}{\lambda_\ell(\lambda_\ell + 1)} \left( \frac{n}{|E|}\lambda_\ell - 1 \right) \sum_{i,j=1}^{n} \left( v_\ell^2(i) + v_\ell^2(j) - 2v_\ell(i)v_\ell(j) \right).$$

We now use that $v_\ell$ form an orthonormal basis, accordingly: $\sum_{i=1}^n v_\ell(i)^2 = 1$. Similarly, since $v_\ell$ is orthogonal to $v_0$ and hence to $\mathbf{1}$, we get $v_\ell^\top \mathbf{1} = \sum_{i=1}^n v_\ell(i) = 0$. Therefore,

$$\sum_{i,j=1}^n \left( v_\ell^2(i) + v_\ell^2(j) - 2v_\ell(i)v_\ell(j) \right) = 2n - 2 \sum_{i=1}^n v_\ell(i) \sum_{j=1}^n v_\ell(j) = 2n.$$

We finally obtain:

$$\frac{1}{n^2} \sum_{i,j=1}^n (\tau_{\mathsf{vn}}(i,j) - \tau(i,j)) = \frac{4|E|}{n} \sum_{\ell=1}^{n-1} \frac{1}{\lambda_\ell(\lambda_\ell + 1)} \left( \frac{n}{|E|} \lambda_\ell - 1 \right),$$

which completes the proof. $\qquad\square$

**Remark:** *We note that according to Equation ($5$), we expect $\tau_{\mathsf{vn}}(i,j) < \tau(i,j)$ whenever the two nodes have large commute time on the input graph, i.e. due to the graph having a bottleneck where $\lambda_1$ is small and the Fiedler eigenvector is such that $v_1(i) \cdot v_i(j) < 0$ since $i, j$ belong to different communities. As such, adding a VN should indeed beneficial for graphs with bottlenecks, while it may slightly increase the commute time between nodes that were well connected in the graph due to the addition of a VN spreading information out from the pair. This intuition is validated in Section $5.1$, where we show that on real-world graphs adding a VN decreases the average commute time significantly.*

**Corollary E.1.** *If we let $\alpha := 1 + n/|E|$, then*

$$-\frac{4|E|\alpha}{\lambda_1(\lambda_1 + 1)} \le \tau_{\mathsf{vn}}(i,j) - \alpha\tau(i,j) \le -\frac{4|E|\alpha}{\lambda_{n-1}(\lambda_{n-1} + 1)}. \tag{35}$$

*Proof.* From ($5$) and the proof above, we derive

$$\tau_{\mathsf{vn}}(i,j) - \alpha\tau(i,j) = \sum_{\ell=1}^{n-1} \left( \frac{2(|E| + n)}{\lambda_\ell + 1} - \frac{2(|E| + n)}{\lambda_\ell} \right) (v_\ell(i) - v_\ell(j))^2$$

$$= -2(|E| + n) \sum_{\ell=1}^{n-1} \frac{1}{\lambda_\ell(\lambda_\ell + 1)} (v_\ell(i) - v_\ell(j))^2$$

$$\ge -\frac{2|E|\alpha}{\lambda_1(\lambda_1 + 1)} \sum_{\ell=1}^{n-1} (v_\ell(i) - v_\ell(j))^2 = -\frac{4|E|\alpha}{\lambda_1(\lambda_1 + 1)}$$

where the last equality follows from

$$\sum_{\ell=1}^{n-1} (v_\ell(i) - v_\ell(j))^2 = \sum_{\ell=1}^{n-1} (v_\ell^2(i) + v_\ell^2(j) - 2v_\ell(i)v_\ell(j))$$

$$= 2\left(1 - \frac{1}{n}\right) - 2 \sum_{\ell=0}^{n-1} v_\ell(i)v_\ell(j) + 2v_0(i)v_0(j) = 2\left(1 - \frac{1}{n}\right) + \frac{2}{n} = 2,$$

where we have used the orthonormality of the eigenvectors for $i \ne j$. The upper bound can be proved in the same way using that $\lambda_\ell \le \lambda_{n-1}$ for each $\ell$. $\qquad\square$

*Proof of Corollary $3.4$.* Consider an MPNN + VN. We recall that $\mathsf{mix}_y(i,j)$ is the mixing induced by a graph-level function among the features of nodes $i$ and $j$, and is defined as (Di Giovanni et al., 2024)

$$\mathsf{mix}_y(i,j) = \max_{\mathbf{H}} \max_{1 \le \alpha, \beta \le d} \left| \frac{\partial^2 y(\mathbf{H})}{\partial h_i^\alpha \partial h_j^\beta} \right|.$$

According to Di Giovanni et al. (2024), when the mixing induced by the MPNN after $m$ layers, denoted here as $\mathsf{mix}^{(m)}(i,j)$, is smaller than the one required by the downstream task, then we have an instance of harmful oversquashing which prevents the network from learning the right interactions

necessary to solve the problem at hand. Since we can write the system in (3) as the system of MPNN analyzed in Di Giovanni et al. (2024), operating on the graph $G_{\mathsf{vn}}$, we can extend their conclusions in Theorem 4.4. Namely, using their notations, we find that given bounded weights and derivatives for their network, the minimal number of layers required to learn a graph-level function $y$ with mixing $\mathsf{mix}_y(i,j)$ is

$$m \geq \frac{\tau_{\mathsf{vn}}(i,j)}{8} + \alpha_{\mathsf{vn}}\mathsf{mix}_y(i,j) + \beta_{\mathsf{vn}},$$

where the constants $\alpha_{\mathsf{vn}}$ and $\beta_{\mathsf{vn}}$ are independent of nodes $i,j$, and without loss of generality we have taken $c_2 = 1/2$ in Theorem 4.4 of Di Giovanni et al. (2024)— clearly, the result generalize to any $c_2 \leq 1$. As such, we see that a necessary condition for the family of graph-level functions with mixing $\mathsf{mix}_y(i,j) > -\beta_{\mathsf{vn}}/\alpha_{\mathsf{vn}}$ to be learned by MPNN + VN is that

$$m \geq \frac{\tau(i,j)}{8} \geq \Big(1 + \frac{n}{|E|}\Big)\Big(\frac{\tau(i,j)}{8} - \frac{|E|}{2\lambda_1(\lambda_1 + 1)}\Big),$$

where we have used Equation (5) in the last inequality. This completes the proof. □

Accordingly, MPNN + VN can improve upon the minimal number of layers required to learn functions with strong mixing between nodes $(i,j)$, when compared to the baseline MPNN. In fact, such improvement can be characterized precisely in terms of spectral properties of the graph Laplacian, and in light of Corollary E.1 of $\lambda_1$.

## F    PROOFS OF RESULTS IN SECTION 4

We begin by providing the analytical model equations for the GatedGCN+VN model that we experiment with in Section 5.

$$h_{\mathsf{vn}}^{(\ell+1)} = \sigma\Big(\mathbf{\Omega}_{\mathsf{vn}}^{(\ell)} h_{\mathsf{vn}}^{(\ell)} + \frac{1}{n}\sum_{j=1}^{n} \mathbf{W}_{\mathsf{vn}}^{(\ell)} h_j^{(\ell)}\Big),$$

$$h_i^{(\ell+1)} = \sigma\Big(\mathbf{\Omega}^{(\ell)} h_i^{(\ell)} + \sum_{j \in N_i} \eta^{(l)}(h_i^{(\ell)}, h_j^{(\ell)}) \odot \mathbf{W}_1^{(\ell)} h_j^{(\ell)} + h_{\mathsf{vn}}^{(\ell)}\Big),$$

where $\odot$ denotes the element-wise product and $\eta^{(l)}(h_i^{(\ell)}, h_j^{(\ell)}) = \mathsf{sigmoid}(\mathbf{W}_2^{(\ell)} h_i^{(\ell)} + \mathbf{W}_3^{(\ell)} h_j^{(\ell)})$. And correspondingly we now write out the analytical model equations for the GatedGCN+VN$_G$ model.

$$h_{j,\mathsf{loc}}^{(\ell+1)} = \sigma(\mathbf{\Omega}^{(\ell)} h_i^{(\ell)} + \sum_{j \in N_i} \eta^{(l)}(h_i^{(\ell)}, h_j^{(\ell)}) \odot \mathbf{W}_1^{(\ell)} h_j^{(\ell)}),$$

$$h_i^{(\ell+1)} = h_{i,\mathsf{loc}}^{(\ell+1)} + \mathsf{Mean}(\{\mathbf{Q}^{(\ell+1)} h_{j,\mathsf{loc}}^{(\ell+1)}\}),$$

where $\eta^{(l)}(h_i^{(\ell)}, h_j^{(\ell)}) = \mathsf{sigmoid}(\mathbf{W}_2^{(\ell)} h_i^{(\ell)} + \mathbf{W}_3^{(\ell)} h_j^{(\ell)})$.

We now continue with the proof of the theoretical statements made in Section 4.

*Proof of Proposition 4.1.* Recall that we consider an MPNN + VN whose layer update is of the form

$$h_{\mathsf{vn}}^{(\ell+1)} = \sigma\Big(\mathbf{\Omega}_{\mathsf{vn}} h_{\mathsf{vn}}^{(\ell)} + \mathbf{W}_{\mathsf{vn}}^{(\ell)} \frac{1}{n}\sum_{j=1}^{n} h_j^{(\ell)}\Big)$$

$$h_i^{(\ell+1)} = \sigma\Big(\mathbf{\Omega}^{(\ell)} h_i^{(\ell)} + \sum_{j=1}^{n} \mathsf{A}_{ij}\psi^{(\ell)}(h_i^{(\ell)}, h_j^{(\ell)}) + \psi_{\mathsf{vn}}^{(\ell)}(h_i^{(\ell)}, h_{\mathsf{vn}}^{(\ell)})\Big).$$

If $i, k \in V$ with $k$ outside the 2-hop neighborhood of $i$, then any message sent from $k$ to $i$ will arrive after 2 layers through the VN. Since the VN update is homogeneous though, node $i$ cannot distinguish which node $k$ sent the message. Explicitly, we can compute the derivatives and obtain

$$\frac{\partial h_i^{(\ell+1)}}{\partial h_k^{(\ell-1)}} = \frac{1}{n}\sigma'(z_i^{(\ell)})\nabla_2\psi_{\mathsf{vn}}^{(\ell)}(h_i^{(\ell)}, h_{\mathsf{vn}}^{(\ell)})\sigma'(z_{\mathsf{vn}}^{(\ell-1)})\mathbf{W}_{\mathsf{vn}}^{(\ell)},$$

where $z_i^{(\ell)}$ is the argument of the nonlinear activation $\sigma$, $\sigma'(z)$ is the diagonal derivative matrix computed at $z$ and $\nabla_2$ denotes the Jacobian w.r.t. the second variable. As we can see, the derivative is independent of $k$ and is the same for each node $k$ outside the 2-hop neighborhood of $i$, which shows that the layer of MPNN + VN is typically homogeneous and hence fails to assign different relevance (sensitivity) to nodes. This completes the proof. □

*Proof of Proposition 4.2.* Consider an instance of MPNN + VN$_G$ whose layer updates have the form

$$h_{i,\mathrm{loc}}^{(\ell+1)} = \sigma(\mathbf{\Omega}^{(\ell)}h_i^{(\ell)} + \sum_{j\in N_i}\psi_{ij}^{(\ell)}(h_i^{(\ell)}, h_j^{(\ell)})),$$

$$h_i^{(\ell+1)} = h_{i,\mathrm{loc}}^{(\ell+1)} + \mathsf{Mean}(\{h_{j,\mathrm{loc}}^{(\ell+1)}\}).$$

Given $i, k \in V$ where node $k$ is outside the 1-hop neighborhood of $i$, any message sent from $k$ to $i$ will arrive after 1 layer through the VN—in other words, $\partial h_{i,\mathrm{loc}}^{(\ell+1)}/\partial h_k^{(\ell)} = 0$. We can compute the derivatives and obtain

$$\frac{\partial h_i^{(\ell+1)}}{\partial h_k^{(\ell)}} = \frac{1}{n}\sum_{j=1}^n \frac{\partial h_{j,\mathrm{loc}}^{(\ell+1)}}{\partial h_k^{(\ell)}}$$

$$= \frac{1}{n}\sum_{j=1}^n \sigma'(z_j^{(\ell)})\Big(\mathbf{\Omega}^{(\ell)}\delta_{jk} + \sum_{u\in N_j}\nabla_1\psi_{ju}(h_j^{(\ell)}, h_u^{(\ell)})\delta_{jk} + \nabla_2\psi_{ju}(h_j^{(\ell)}, h_u^{(\ell)})\delta_{uk}\Big)$$

$$= \frac{1}{n}\Big(\sigma'(z_k^{(\ell)})\Big(\mathbf{\Omega}^{(\ell)} + \sum_{u\in N_k}\nabla_1\psi_{ku}^{(\ell)}(h_k^{(\ell)}, h_u^{(\ell)})\Big) + \sum_{u\in N_k}\sigma'(z_u^{(\ell)})\nabla_2\psi_{uk}^{(\ell)}(h_u^{(\ell)}, h_k^{(\ell)})\Big),$$

where $\delta_{pq}$ is the Kronecker delta with indices $p, q$ and in the last sum we have replaced the dumb index $j$ with $u$. This completes the proof. □

For completeness, we also provide a corresponding sensitivity analysis of the global attention mechanism now, that is at the heart of the different GT variants. In particular, we will be analysing the following global attention mechanism,

$$\mathbf{H}_{\mathrm{att}}^{(\ell+1)} = \mathrm{softmax}\left(\frac{1}{\sqrt{d_{\ell+1}}}\mathbf{H}^{(\ell)}\mathbf{W}_{\mathrm{Q}}^{(\ell)}(\mathbf{H}^{(\ell)}\mathbf{W}_{\mathrm{K}}^{(\ell)})^\top\right)\mathbf{H}^{(\ell)}\mathbf{W}_{\mathrm{V}}^{(\ell)}, \tag{36}$$

where $\mathbf{W}_{\mathrm{Q}}^{(\ell)}, \mathbf{W}_{\mathrm{K}}^{(\ell)}, \mathbf{W}_{\mathrm{V}}^{(\ell)}$ denote trainable weight matrices and $d_{\ell+1}$ denotes the number of columns of $\mathbf{W}_{\mathrm{Q}}^{(\ell)}$.

**Proposition F.1.** *Given $i, k \in V$ the Jacobian $\partial h_i^{(\ell+1)}/\partial h_k^{(\ell)}$ computed using (36) is*

$$\frac{\partial h_i^{(\ell+1)}}{\partial h_k^{(\ell)}} = \mathrm{softmax}\left(\frac{1}{\sqrt{d_{\ell+1}}}(h_i^{(\ell)})^\top\mathbf{W}_{\mathrm{Q}}^{(\ell)}(\mathbf{W}_{\mathrm{K}}^{(\ell)})^\top h_k^{(\ell)}\right)\mathbf{W}_{\mathrm{V}}^{(\ell)}$$

$$+ \mathrm{softmax}'\left(\frac{1}{\sqrt{d_{\ell+1}}}(h_i^{(\ell)})^\top\mathbf{W}_{\mathrm{Q}}^{(\ell)}(\mathbf{W}_{\mathrm{K}}^{(\ell)})^\top h_k^{(\ell)}\right)\frac{1}{\sqrt{d_{\ell+1}}}(h_i^{(\ell)})^\top\mathbf{W}_{\mathrm{Q}}^{(\ell)}(\mathbf{W}_{\mathrm{K}}^{(\ell)})^\top\mathbf{W}_{\mathrm{V}}^{(\ell)}h_k^{(\ell)},$$

*Proof.* We begin by reformulating the global attention mechanism equation in (36) to reflect the update of the hidden representation $h_{\mathrm{att},i}^{(\ell+1)}$ of node $i$,

$$h_{\mathrm{att},i}^{(\ell+1)} = \sum_{j=1}^n \mathrm{softmax}\left(\frac{1}{\sqrt{d_{\ell+1}}}(h_i^{(\ell)})^\top\mathbf{W}_{\mathrm{Q}}^{(\ell)}(\mathbf{W}_{\mathrm{K}}^{(\ell)})^\top h_j^{(\ell)}\right)\mathbf{W}_{\mathrm{V}}^{(\ell)}h_j^{(\ell)}.$$

Now the Jacobian can be relatively simply derived using the product rule as follows,

$$
\frac{\partial h_i^{(\ell+1)}}{\partial h_k^{(\ell)}} = \mathrm{softmax}\left(\frac{1}{\sqrt{d_{\ell+1}}}(h_i^{(\ell)})^\top \mathbf{W}_Q^{(\ell)}(\mathbf{W}_K^{(\ell)})^\top h_k^{(\ell)}\right)\mathbf{W}_V^{(\ell)}
$$
$$
+ \mathrm{softmax}'\left(\frac{1}{\sqrt{d_{\ell+1}}}(h_i^{(\ell)})^\top \mathbf{W}_Q^{(\ell)}(\mathbf{W}_K^{(\ell)})^\top h_k^{(\ell)}\right)\frac{1}{\sqrt{d_{\ell+1}}}(h_i^{(\ell)})^\top \mathbf{W}_Q^{(\ell)}(\mathbf{W}_K^{(\ell)})^\top \mathbf{W}_V^{(\ell)} h_k^{(\ell)},
$$

where $\mathrm{softmax}'$ denotes the derivative of the $\mathrm{softmax}(\cdot)$ function. $\qquad\square$

We can hence conclude that hidden states obtained in the global attention mechanism, that is used in most GTs without modification, depend on both the hidden state of the central node $h_i^{(\ell)}$ and the hidden state of any other given node $h_k^{(\ell)}$. This dependence arises rather trivially as a result of the fully connected attention scheme, in which any two nodes can exchange information.

## G  EXPERIMENTAL DETAILS

In this section, we provide further details about our experiments.

### G.1  HARDWARE

All experiments were run on a single V100 GPU.

### G.2  DESCRIPTION OF DATASETS

Below, we provide descriptions of the datasets on which we conduct experiments.

**MNIST and CIFAR10** (CC BY-SA 3.0 and MIT License) (Dwivedi et al., 2023). These datasets test graph classification on the popular image classification datasets. The original images are converted to graphs using super-pixels which represent small regions of homogeneous intensity in the images. They are both 10-class classification tasks and follow the original standard (train/validation/test) splits; 55K/5K/10K for MNIST and 45K/5K/10K for CIFAR10.

**ogbg-molhiv and ogbg-molpcba** (MIT License) (Hu et al., 2020). These are molecular property prediction datasets which use a common node and edge featurization that represents chemophysical properties. Ogbg-molhiv is a binary classification task, predicting the molecule's ability to inhibit HIV replication and ogbg-molpcba is a multi-task binary classification where 128 bioassays are predicted.

**ogbg-ppa** (CC-0 License) (Hu et al., 2020). This dataset consists of protein-protein interaction networks derived from 1581 species and categorized into 37 taxonomic groups. Nodes represent proteins and edges encode the normalized level of 7 different associations between proteins. The task is to classify which of the 37 groups the network belongs to.

**MalNet-Tiny** (CC-BY License) (Freitas et al., 2020). This is a subset of MalNet which contains function call graphs (FCGs) derived from Android APKs. There are 5,000 graphs with up to 5,000 nodes with each graph coming from a benign software. The goal is to predict the type of software from the structure of the FCG. The benchmarking version of this dataset typically uses Local Degree Profile as the set of node features.

**Peptides-func and Peptides** (CC-BY-NC 4.0) (Dwivedi et al., 2022). These datasets are composed of atomic peptides. Ppetides-func is a multi-label graph classification task where there are 10 nonexclusive peptide functional classes. Peptides-struct is a regression task involving 11 3D structural properties of the peptides.

**Sparsity of Graphs and Complexity of VN.** The real-world benchmarks used in this paper are generally very sparse. For instance, ogbg-molhiv (mean number of nodes: 25.5 , mean number of edges: 27.5), Peptides (mean number of nodes: 150.9 , mean number of edges: 307.3), MNIST (mean number of nodes: 70.6 , mean number of edges: 564.5), CIFAR10 (mean number of nodes: 117.6 , mean number of edges: 941.1), MalNet-Tiny (mean number of nodes: 1,410.3 , mean number

of edges: 2,859.9). The computational complexity of an MPNN is $O(|E|)$ and of an MPNN+VN is $O(|E| + n)$. Given that, on these datasets, the order of magnitude of $n$ is similar to $|E|$, the complexity of MPNN+VN can be written as $O(cn)$ where $c$ is a small constant. This is significantly better than the computational complexity of a Graph Transformer which is $O(n^2)$ on these datasets.

## G.3 DATASET SPLITS AND RANDOM SEEDS

All the benchmarks follow the standard train/validation/test splits. The test performance at the epoch with the best validation performance is reported and is averaged over multiple runs with different random seeds. All the benchmarking results, including the extra ablations, are based on 10 executed runs, except for Peptides-func and Peptides-struct which are based on the output of four runs.

## G.4 HYPERPARAMETERS

Considering the large number of hyperparameters and datasets, it was not possible to do an exhaustive grid search and to find the optimal parameters. Here we describe how the final hyperparameters shown in Tables 4 and 5 were obtained.

**Hyperparameters in Table 1**. For the GPS model and its projection, we used the hyperparameters as described in the original work Rampášek et al. (2022). For GatedGCN+PE+VN and other trained models, we outline our hyperparameter optimization process for different datasets in the following subsections.

**OGB datasets**. For both ogbg-molpcba and ogbg-ppa we used the same hyperparameters as used in Rampášek et al. (2022) but the hidden dimension of the MPNN was adjusted so that we had a similar parameter budget. For ogbg-molhiv, we found it to be beneficial to reduce the number of layers to 4 to align with the number of layers used in Bouritsas et al. (2023) but kept the same parameters for the positional encoding and the optimization process.

**Peptides-Func and Peptides-Struct**. For these datasets we optimized the hyperparameters over the following ranges:

- Dropout [0, 0.1, 0.2],
- FeedForward Block [True, False],
- Depth [4, 6, 8, 10],
- Positional Encoding [none, LapPE, RWSE],
- Layers Post Message-Passing [1, 2, 3],

and we used the optimization parameters recently proposed by (Tönshoff et al., 2023a) where they train for 250 epochs with an AdamW optimizer (Kingma & Ba, 2014), and a cosine annealing learning rate scheduler with a base learning rate of 0.001. When optimizing over the number of layers of message-passing, we changed the hidden dimension to ensure that the parameter budget was around 500K.

**MNIST, CIFAR10, MalNet-Tiny**. On these benchmarks, we used the same dropout, positional encodings and optimization parameters as used by Shirzad et al. (2023). The only parameter we optimised for was the number of layers in the range [3, 5, 7]. Additionally, we changed the hidden dimension in accordance with the number of layers to match the number of parameters that were used in Shirzad et al. (2023).

Table 4: Best performing hyperparameters for GatedGCN+PE+VN$_G$ in Table 2.

| Hyperparameter | Peptides-Func | Peptides-Struct | MNIST | CIFAR10 | MalNet-Tiny |
|---|---|---|---|---|---|
| #Layers | 4 | 4 | 5 | 7 | 5 |
| Hidden dim | 136 | 136 | 46 | 40 | 72 |
| Dropout | 0.0 | 0.2 | 0.1 | 0.1 | 0.0 |
| Graph pooling | mean | mean | mean | mean | mean |
| FeedForward Block | False | False | True | True | True |
| Positional Encoding | RWSE-20 | LapPE-16 | LapPE-8 | LapPE-8 | None |
| PE dim | 16 | 16 | 8 | 8 | - |
| PE encoder | Linear | Linear | Linear | DeepSet | - |
| Batch size | 200 | 200 | 16 | 16 | 16 |
| Learning Rate | 0.001 | 0.001 | 0.001 | 0.001 | 0.0005 |
| #Epochs | 250 | 250 | 100 | 100 | 150 |
| #Parameters | 492,217 | 492,217 | 110,148 | 117,066 | 281,453 |

Table 5: Best performing hyperparameters for GatedGCN+PE+VN$_G$ in Table 3.

| Hyperparameter | ogbg-molhiv | ogbg-molpcba | ogbg-ppa |
|---|---|---|---|
| #Layers | 4 | 5 | 3 |
| Hidden dim | 90 | 384 | 284 |
| Dropout | 0.0 | 0.2 | 0.1 |
| Graph pooling | mean | mean | mean |
| FeedForward Block | True | True | True |
| Positional Encoding | RWSE-16 | RWSE-16 | None |
| PE dim | 16 | 20 | - |
| PE encoder | Linear | Linear | - |
| Batch size | 32 | 512 | 32 |
| Learning Rate | 0.0001 | 0.0005 | 0.0003 |
| #Epochs | 100 | 100 | 200 |
| #Parameters | 360,221 | 7,519,580 | 2,526,501 |

# H   SMOOTHING AND THE IMPORTANCE OF GRAPH STRUCTURE

In this section, we explore the performance on various benchmarks of adding or removing the current pooled representation at each layer. We take the pooling function to be the mean and at each layer we either subtract or add this from the output of the message passing layer. Adding the mean increases the smoothness of the representations and we see that this leads to performance improvements on the LRGB tasks in Table 6. Again, this highlights the benefits of smoothing on some graph-level tasks. On the other hand, we see that subtracting the mean is beneficial for ogbg-molpcba. Subtracting the mean at each layer before using the mean pooling function in the final layer, means that we are ignoring the output of the message-passing layer which is aligned with the final representation. We can see this as a good measure of the importance of the graph structure as we are removing any locally aggregated updates. This suggests that it is beneficial to ignore the graph topology for ogbg-molpcba.

Table 6: Analyzing the performance of GatedGCN where we subtract or add the mean of the message-passing output at each layer.

| Method | Peptides-Func ($\uparrow$) | Peptides-Struct ($\downarrow$) | ogbg-molhiv ($\uparrow$) | ogbg-molpcba ($\uparrow$) |
|---|---|---|---|---|
| GatedGCN | 0.5864 $\pm$0.0077 | 0.3420 $\pm$0.0013 | 0.7827 $\pm$0.0111 | 0.2714 $\pm$0.0014 |
| GatedGCN + mean | 0.6692 $\pm$0.0042 | 0.2522 $\pm$0.0012 | 0.7677 $\pm$0.0138 | 0.2569 $\pm$0.0034 |
| GatedGCN - mean | 0.4675 $\pm$0.0040 | 0.3566 $\pm$0.0007 | 0.7594 $\pm$0.0096 | 0.2866 $\pm$0.0016 |

## H.1   EMPIRICAL COMPARISON OF MPNN + VN AND PAIRNORM

We now study the question *to what extent* MPNN + VN *replicates PairNorm on graph-level tasks.* To answer this question and assess whether MPNN + VN behaves differently to PairNorm in practice, we evaluate both on two graph-level tasks from the Long-Range Benchmarks (Dwivedi et al., 2022) using three different MPNN architectures. It is clear from Table 7 that whilst PairNorm has a damaging impact on the results as hypothesized in Theorem B.1, MPNN + VN achieves significant gains over the standard MPNN architecture. Further datasets and results using a GatedGCN can be found in

Table 9. One explanation for the observed phenomenon is that, on these datasets, MPNN + VN favors alignment between the layerwise representations and the final output representation (a smoothing process). This phenomenon can be observed when comparing the cosine similarity of the pooled representations at each layer to the final pooled representation. Plots of this cosine similarity over the layers can be seen in Appendix H.2 and clearly show that the VN causes a smoothing effect in contrast to PairNorm.

Table 7: The effect of PairNorm and MPNN + VN on LRGB tasks.

| Method | Peptides-Func ($\uparrow$) | Peptides-Struct ($\downarrow$) |
|---|---|---|
| GCN | 0.5930 $\pm$0.0023 | 0.3496 $\pm$0.0013 |
| GCN + PairNorm | 0.4980 $\pm$0.0031 | 0.3471 $\pm$0.0016 |
| GCN + VN | **0.6623** $\pm$0.0038 | **0.2488** $\pm$0.0021 |
| GINE | 0.5498 $\pm$0.0079 | 0.3547 $\pm$0.0045 |
| GINE + PairNorm | 0.4698 $\pm$0.0053 | 0.3562 $\pm$0.0007 |
| GINE + VN | **0.6346** $\pm$0.0071 | **0.2584** $\pm$0.0011 |
| GatedGCN | 0.5864 $\pm$0.0077 | 0.3420 $\pm$0.0013 |
| GatedGCN + PairNorm | 0.4674 $\pm$0.0040 | 0.3551 $\pm$0.0008 |
| GatedGCN + VN | **0.6477** $\pm$0.0039 | **0.2523** $\pm$0.0016 |

As argued before, this further supports that on these tasks, the alignment between the VN and the choice of global pooling means that the task depends on feature information associated with the frequency components of the subspace spanned by $\mathbf{1}$.

## H.2 COSINE DISTANCE PLOTS

We analyzed the layerwise smoothing on the graph-level LRGB datasets (Dwivedi et al., 2022) using a GatedGCN with and without a VN/attention layer. To do this, we measured the cosine distance between the representations at each layer and the final pooled graph representation. We found that using a VN or an attention layer reduced the distance between the earlier layer representations and the final pooled representation and that this lead to an improvement in performance. This suggests that these approaches can cause a beneficial smoothing towards the final representation.

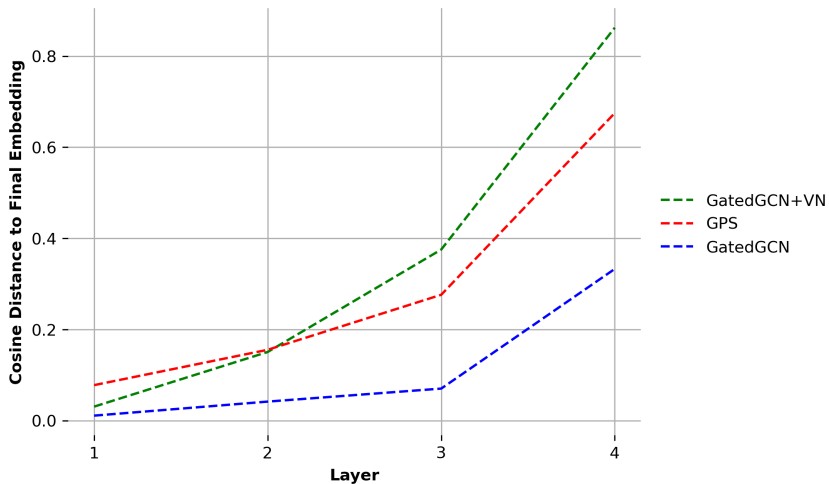

Figure 2: The cosine distance between the embedding at each layer and the final graph representation after training a GatedGCN, as well as with a VN and a transformer layer on Peptides-func.

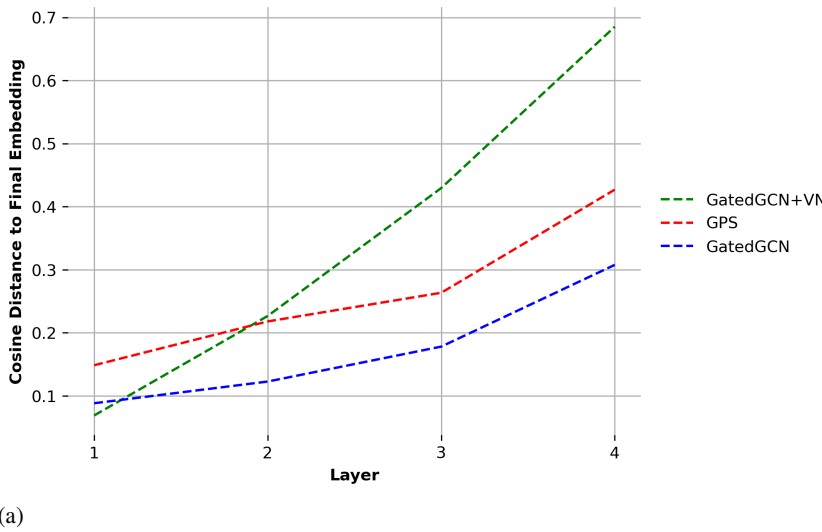

(a)

Figure 3: The cosine distance between the embedding at each layer and the final graph representation after training a GatedGCN, as well as with a VN and a transformer layer on Peptides-struct.

## I   ADDITIONAL ABLATION STUDIES

In this section, we explore additional ablations to improve our understanding of virtual nodes and our heterogeneous extension. In Table 8, we look at the performance improvement of our heterogeneous $VN_G$ when the base MPNN is a GCN or a GatedGCN. Whilst we generally see an improvement with our implementation, this improvement is much larger when we use a GatedGCN. As previously argued, this is due to the fact that in a GatedGCN, the aggregation weights each neighbor through a learnable gate and we can thus have a learnable node importance based on the graph topology. This is in contrast to a GCN where we have a homogeneous node update.

Table 8: Performance of MPNN + $VN_G$ in comparison to virtual node with positional encodings.

| Method | ogbg-molhiv (↑) | ogbg-molpcba (↑) | peptides-func (↑) | peptides-struct (↓) | CIFAR10 (↑) |
|---|---|---|---|---|---|
| GCN+PE+VN | 0.7599 ±0.0119 | 0.2456 ±0.0034 | 0.6732 ±0.0068 | 0.2475 ±0.0009 | 68.957 ±0.381 |
| GCN+PE+$VN_G$ | 0.7678 ±0.0111 | 0.2481 ±0.0032 | 0.6862 ±0.0023 | 0.2456 ±0.0010 | 68.756 ±0.172 |
| GatedGCN+PE+VN | 0.7687 ±0.0136 | 0.2848 ±0.0026 | 0.6712 ±0.0066 | 0.2481 ±0.0015 | 70.280 ±0.380 |
| GatedGCN+PE+$VN_G$ | 0.7884 ±0.0099 | 0.2917 ±0.0027 | 0.6822 ±0.0052 | 0.2458 ±0.0006 | 76.080 ±0.301 |
| GCN % Increase | +1.04 | +1.02 | + 1.93 | +0.77 | -0.29 |
| GatedGCN % Increase | +2.56 | +2.42 | +1.64 | +0.89 | +8.25 |

As an extension to Section B, we compared the performance of GatedGCN with augmentations which involve adding a VN in Table 9. Additionally, we look at the effect of applying PairNorm on the datasets. We see that applying PairNorm to the GatedGCN, a common technique to mitigate oversmoothing, actually reduces performance on all of these datasets. Moreover, using a virtual node always outperforms using PairNorm. This further implies that a VN is not recreating PairNorm and suggests that, on these graph-level tasks, smoothing may be beneficial.

Table 9: Performance of GatedGCN and its extensions on four benchmark tasks.

| Method | Peptides-Func (↑) | Peptides-Struct (↓) | ogbg-molhiv (↑) | ogbg-molpcba (↑) |
|---|---|---|---|---|
| GatedGCN | 0.5864 ±0.0077 | 0.3420 ±0.0013 | 0.7827 ±0.0111 | 0.2714 ±0.0014 |
| GatedGCN with PairNorm | 0.4674 ±0.0040 | 0.3551 ±0.0008 | 0.7645 ±0.0128 | 0.2621 ±0.0026 |
| GatedGCN + VN | 0.6477 ±0.0039 | 0.2523 ±0.0016 | 0.7676 ±0.0172 | 0.2823 ±0.0026 |

## J   VISUALIZATION OF MPNN + $VN_G$

Here we visualize the differences between our implementation of MPNN + $VN_G$ and a standard MPNN+VN. Our model uses the local MPNN acting on the original graph to weight the nodes based on their importance. This means that the VN can perform a heterogeneous global update.

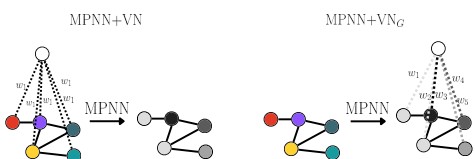

Figure 4: Comparing MPNN+VN with our proposed MPNN + $VN_G$.

## K    ATTENTION MAPS

In order to get a better understanding of the homogeneity of the attention layer in the GPS framework, we visualized the first layer attention matrices for various datasets. This complements our analysis in Section 4 where we relate the gap in performance between a MPNN + VN and GPS to the level of homogeneity of these attention patterns. The attention pattern for CIFAR10 is less homogeneous, as measured by the standard deviation of the columns sums.

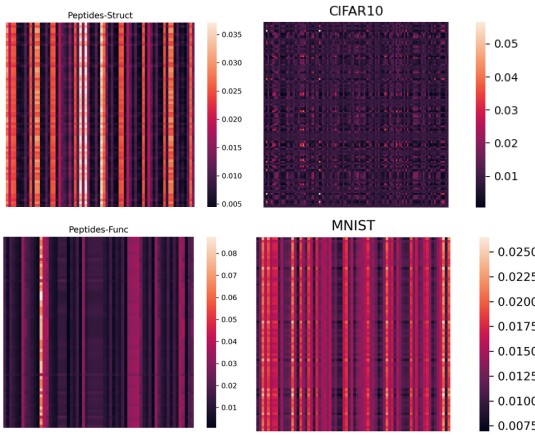

Figure 5: First layer attention maps of the self-attention matrix in the GPS framework for different datasets.

