# OpenReview forum: "Understanding Virtual Nodes: Oversquashing and Node Heterogeneity"
_ICLR.cc/2025/Conference — ICLR 2025 Poster_

### Official Review · Reviewer_kkhT · 2024-10-31

**Soundness:** 3
**Presentation:** 3
**Contribution:** 2
**Rating:** 5
**Confidence:** 4

**Summary:**

This paper provides a comprehensive theoretical analysis of the role of virtual nodes in message passing neural networks, specifically addressing the oversquashing problem and long-range interaction limitations. The authors characterize how virtual nodes improve network mixing abilities and mitigate oversquashing based on underlying network topology. Additionally, they propose a variant of virtual nodes that can assign different importance to nodes based on graph structure, offering a more effective and computationally efficient solution for graph-level tasks compared to traditional approaches.

**Strengths:**

This paper provides a detailed theoretical analysis addressing the issues of "oversquashing and node heterogeneity".
Each chapter of this paper presents a detailed problem statement, followed by a comprehensive theoretical analysis and the design of corresponding algorithms to tackle these issues.

**Weaknesses:**

1.Although this paper provides a lot of theoretical analysis and derivation, the conclusions are unclear. For example, Theorem 3.1. The conclusion is a complex expression. Can the author further simplify it through approximation or other means?

2.The experimental results of this paper show almost no improvement compared to other methods (Tables 1-3).

3.Although the content of this paper is substantial, it lacks a framework diagram. Readers find it difficult to intuitively understand what contributions this paper has and the relationships between the various contributions.

4.This paper attempts to highlight its experimental results through bright colors, but the bright green color makes readers uncomfortable. In addition, the excessive use of different colors makes the experimental results appear somewhat messy, for example in Table 2.

**Questions:**

The MNIST and CIFAR10 datasets are both image datasets. Why are they being used in graph-related research or this paper?

---

> ### Author Response · Authors · 2024-11-20
> **Response to Reviewer kkhT**
>
> We thank the reviewer for their feedback and their response. Here, we respond to the weaknesses and questions point by point.
>
> _**"Although this paper provides a lot of theoretical analysis and derivation, the conclusions are unclear. For example, Theorem 3.1. The conclusion is a complex expression. Can the author further simplify it through approximation or other means?"**_
>
> We do not see a clear way to further simplify the expression in Theorem 3.1. In general, we believe it is important to give an exact statement of the result (regardless of complexity) and then to follow this with simple and clear interpretations of such a result. We believe that Section 3 provides this clarity. Theorem 3.1. measures the commute time change for a graph when adding a virtual node. We show that on all datasets used, adding a virtual node reduces the commute time. This is an important conclusion and highlights the benefits of using a virtual node in terms of oversquashing and explains (to an extent) the performance gap between MPNNs and MPNN+VN.
>
> _**"The experimental results of this paper show almost no improvement compared to other methods (Tables 1-3)."**_
>
> We respectfully disagree. We significantly improve over a standard MPNN+VN, with the same computational cost, on **all** tasks. MPNN+VN is a popular and practical baseline often used in the community because of the simplicity of these architectures. Additionally, we get the top performing results on 4/8 tasks and the second best performing on 3/8 tasks. Given that we are comparing to some of the SOTA approaches, including GTs which are more computationally expensive, we believe that these results are certainly significant and interesting to the community.
>
> _**"Although the content of this paper is substantial, it lacks a framework diagram. Readers find it difficult to intuitively understand what contributions this paper has and the relationships between the various contributions."**_
>
> We thank the reviewer for bringing this to our attention. We have listed our main contributions as clearly as possible in lines 60-70 of the main paper. These include 1) Studying the impact of a VN on oversquashing, 2) Comparing VNs and GTs in terms of node sensitivity and 3) Proposing MPNN+VN_G and showcasing its benefit on a range of tasks.
>
> Given the extensive and varying theoretical contributions of the paper, we found it difficult to visualize our contributions with a framework figure and we felt that this would actually reduce clarity. We have included a Figure of the MPNN+VN_G update in Appendix J which we hoped could clarify some of the discussion in Section 4. Do you have a particular idea for a framework diagram? If so, we would be happy to include it in a revision.
>
> _**"This paper attempts to highlight its experimental results through bright colors, but the bright green color makes readers uncomfortable. In addition, the excessive use of different colors makes the experimental results appear somewhat messy, for example in Table 2."**_
>
> We have tried to highlight the top performing results and used the color scheme to provide clarity that our method can achieve substantial performance improvements. There are several publications in the related literature, that use several colours to highlight the top performing models on these datasets, e.g. [2,3,4]. We have now changed the bright green color in the revision to be olive in the hope that it makes the results less uncomfortable to view.
>
> _**"The MNIST and CIFAR10 datasets are both image datasets. Why are they being used in graph-related research or this paper?"**_
>
> The MNIST and CIFAR10 datasets are derived from like-named image classification datasets by constructing an 8 nearest-neighbor graph of SLIC superpixels for each image. These graph datasets (derived from images) have been used extensively by the graph learning community [1,2,3,4] as they provide a substantially different graph structure to commonly used molecular datasets. This means that models can be tested and compared on a wider range of input graphs.
>
> Thanks again for your review and the comments which have helped improve the manuscript. We hope that we have sufficiently addressed your points and concerns. Please let us know if this remains unclear or if you consider any discussion points to be open.
>
> [1] Toenshof et al. Walking Out of the Weisfeiler Leman Hierarchy: Graph Learning Beyond Message Passing. TMLR 2023.
>
> [2] Ma et al. Graph Inductive Biases in Transformers without Message Passing. ICML 2023.
>
> [3] Rampášek et al. Recipe for a General, Powerful, Scalable Graph Transformer. NeurIPS 2022.
>
> [4] Shirzad et al. EXPHORMER: Sparse Transformers for Graphs. ICML 2023.

---

> > ### Author Response · Authors · 2024-11-26
> > **Further discussion would be much appreciated**
> >
> > Dear Reviewer kkhT,
> >
> > Thank you very much for the time you have dedicated to reviewing our paper. Given the extended discussion time, we feel that further correspondence with you would be of great value to the manuscript.
> >
> > We hope that our rebuttal has resolved your concerns on the formatting and that we have sufficiently addressed the comments about the empirical performance and presentation of the theoretical results.
> >
> > Please let us know if any questions remain which substantiate the score currently given for the paper.

---

### Official Review · Reviewer_aG7n · 2024-10-31

**Soundness:** 3
**Presentation:** 4
**Contribution:** 1
**Rating:** 3
**Confidence:** 4

**Summary:**

Message passing neural networks (MPNNs) are known to be limited in their capability to capture long-range interactions between nodes, a problem known as oversquashing. This paper studies the effect of adding a virtual node (VN) on over-squashing, giving an exact characterization in terms of spectral properties of the network.

Oversquashing is the problem of fitting information in a constant-sized embedding vector from a neighborhood growing exponentially in terms of radius. This problem is known to occur between nodes at a high commute time from each other. The paper calculates the change in commute time from adding a virtual node, as a function of eigenvectors of the network Laplacian. The change in commute time is verified empirically.

Additionally, the paper does a brief comparison of MPNN + VN to Graph Transformers (GTs). In light of this comparison, the authors give (formalize) a slightly different VN architecture with interleaved node and VN updates. This architecture is claimed to be better because it incorporates graph structure before aggregating messages at the virtual node. This model is also tested in practice.

**Strengths:**

The paper gives a precise characterization of the difference in commute time when a virtual node is added. Combined with the results on commute time from previous work, this gives a clear insight into how VNs mitigate oversquashing. The difference in commute time is given for pairwise commute time, as well as the average commute time over the network. The result is also connected to the concept of mixing from earlier work, showing that MPNNs with a virtual node can require fewer layers to represent functions that require high mixing.

The experimental results align with the theoretical findings. The paper is also well written and easy to follow.

**Weaknesses:**

The comparison of MPNN + VN to Graph Transformers (Section 4) feels somewhat superficial and does not offer significant new insights. The main observation is that graph transformer layer allows for heterogeneous node importance, while MPNN+VN treats nodes with homogeneous importance.

The authors propose MPNN + VN_G, a formulation with node updates interleaved with the VN update. This formulation is not new, and the authors also say it has been previously shown to improve performance in practice. The claim is that the update allows heterogeneous attention to different nodes, but this seems like a questionable claim. Heterogeneity is due to the different graph topologies: high-degree nodes affect a different number of neighbors proportional to their degree, before new states are aggregated to the VN. This is different from heterogeneity in GTs via the attention mechanism, where different importances can be learned. There is a note about this on line 349, but I feel like the section is still overselling its results. It is also worth mentioning that a traditional MPNN + VN is also capable of heterogeneous node importance in the weaker sense, just with at least 3 layers.

Overall, the primary contribution of the paper seems to be Theorem 3.1. A downside of this theorem is its practical application. The authors mention that the change in commute time from adding a virtual node can sometimes be negative (e.g. in a complete graph). The experiments show significant avereage commute time reduction for some graph datasets, but the paper does not make it clear whether these experimental results can be reliably generalized in practical applications.

Given that commute time was already introduced in previous work, all we have now seems to be a formula. The formula is complicated and without real practical use. Theory is best if it is actually usable, but I don't know how to use this theory.

**Questions:**

While the runtime of MPNN + VN_G is the same as MPNN + VN, could there be problems with vanishing gradients since the depth is essentially doubled?

Minor comment: equation 14: ": j \in V" missing. same below on line 311

---

> ### Author Response · Authors · 2024-11-20
> **Response to Reviewer aG7n**
>
> We thank the reviewer for their feedback and their response. Here, we try to respond to the weaknesses and questions.
>
> _**"The comparison of MPNN + VN to Graph Transformers (Section 4) feels somewhat superficial and does not offer significant new insights."**_
>
> The comparison between two popular architectures for graph learning; MPNN+VN and Graph Transformers (GTs) has led to other works [1][2] and previously been interesting to the community. We extend and build on these works by comparing these methods in terms of heterogeneous sensitivity. We argue that this provides significant new insights. Firstly, it explains why and on what datasets there will be a performance gap between MPNN+VN and GTs (Table 1). Additionally, it provides a framework for improving MPNN+VN by allowing more heterogeneous sensitivity. We believe that substantially improving MPNN+VN, without increasing the computational complexity, and explaining this through sensitivity analysis does provide important insight for the community.
>
> [1] Cai et al. On the Connection Between MPNN and Graph Transformer. ICML 2023.
>
> [2] Rosenbluth et al. Distinguished in Uniform: Self-Attention Vs. Virtual Nodes. ICLR 2024.
>
> _**"The authors propose MPNN + VN_G, a formulation with node updates interleaved with the VN update. This formulation is not new, and the authors also say it has been previously shown to improve performance in practice."**_
>
> Previous works have indeed found that interleaving a global and local update can improve performance on benchmarks [3]. However, no explanation has been given for _**why**_ this is the case. Additionally, the methods are different from our implementation MPNN+VN_G. For example, the global update in [3] is a transformer. We believe that **explaining** the performance improvements and **providing a new method** (with the same computational complexity as MPNN+VN) that performs well across a range of datasets is a significant and novel contribution.
>
> [3] Yin et al.  Graph transformers with local and global operators interleaving. IJCAI 2023.
>
> _**"The claim is that the update allows heterogeneous attention to different nodes, but this seems like a questionable claim. Heterogeneity is due to the different graph topologies: high-degree nodes affect a different number of neighbors proportional to their degree, before new states are aggregated to the VN. This is different from heterogeneity in GTs via the attention mechanism, where different importances can be learned. There is a note about this on line 349, but I feel like the section is still overselling its results."**_
>
> While we agree with the reviewer that the GCN+VN_G assigns heterogeneous node importance based on the graph topology, with high degree nodes having a higher impact. The same cannot be said about the GatedGCN+VN_G model, which is the main baseline we consider in our experiments in Section 5. From the corresponding model equations, provided at the beginning of Appendix F, it is readily apparent that the trainable function $\eta^{(l)}(h_i^{(\ell)}, h_j^{(\ell)})$, is a simple form of attention, that allows the GatedGCN+VN_G model to assign heterogeneous node importance in a trainable manner that is free of topological constraints.
>
> _**"Overall, the primary contribution of the paper seems to be Theorem 3.1. A downside of this theorem is its practical application. The authors mention that the change in commute time from adding a virtual node can sometimes be negative (e.g. in a complete graph)."**_
>
> We agree with the reviewer that Theorem 3.1 is one of the primary contributions of the paper. However, we disagree that it has no practical relevance. For all of the datasets used in the paper, adding a VN reduces the commute time (Figure 1). This can explain the performance improvements seen in practice when using a VN on these datasets. We believe that understanding why a certain method works well has enormous utility. It is true that on some graphs we may not reduce the commute time by adding a VN. We think that this increases the utility of this theorem even more as it can tell us when adding a VN is not beneficial. It is worth noting though that complete and dense graphs are not realistic in practical settings and would be more suited to GTs. In practice, real-world graphs tend to be sparse, as are all the benchmarks studied in the paper, and it is in this case where VN_G is shown to be beneficial.
>
> We also respectfully disagree that this is the only important contribution of the paper and feel that the sensitivity analysis (Section 4) and the smoothing analysis (Appendix B) go a long way in both explaining why MPNN+VN is successful and how we can improve it.

---

> > ### Author Response · Authors · 2024-11-20
> > **Response to Reviewer aG7n, pt 2**
> >
> > _**"The experiments show significant avereage commute time reduction for some graph datasets, but the paper does not make it clear whether these experimental results can be reliably generalized in practical applications."**_
> >
> > Indeed we find a significant reduction in commute time based on Theorem 3.1 on several sparse real-world graph datasets (shown in Figure 1). We have now highlighted the sparsity of the graphs in the benchmarks used in Appendix G.2 and shown that this is also where VNs make the most sense in terms of complexity. In terms of generalizing this finding to practical applications, we believe that Section 5 and our experimental results provide a good attempt at this. Peptides dataset involves predicting functional classes and 3D structural properties of peptide chains, an important practical application in Bioinformatics, and MOLHIV predicts a molecule’s fitness to inhibit HIV replication. We feel that these are real-world datasets drawn from important practical applications which are substantially improved by both reducing the commute time and adding heterogeneous sensitivity in the VN update.
> >
> >
> > _**"Given that commute time was already introduced in previous work, all we have now seems to be a formula. The formula is complicated and without real practical use. Theory is best if it is actually usable, but I don't know how to use this theory."**_
> >
> > We believe it is important to give an exact statement of the theoretical result (regardless of complexity) and then to follow with simple and clear interpretations of such a result. We believe that Section 3 provides this clarity. Theorem 3.1. measures the commute time change for a graph when adding a virtual node. We show that on all datasets used, adding a virtual node reduces the commute time. This is an important conclusion and highlights the benefits of using a virtual node in terms of oversquashing, and explains (to an extent) the performance gap between MPNNs and MPNN+VN.
> > Our theoretical contributions can be seen as providing further understanding to practitioners on the differences between MPNNs vs MPNN+VNs, and MPNN+VNs vs GTs. We show that these theoretical differences explain the performance differences between them, we explain why and on what datasets this is likely to occur, and we use this theoretical framework to substantially improve the MPNN+VN architecture. We find it to be a rather successful example when theory matches well with practice.
> >
> > _**"Minor comment: equation 14: ": j \in V" missing. same below on line 311"**_
> >
> > To us it is slightly unclear where the “$j \in V$” statement may be missing in Equation (14). If we were to specify repeatedly throughout the paper that $i$ and $j$ are in $V$ and denote nodes, this may clutter the paper. But this may not be what you are suggesting. Could you please clarify your suggestion, we would be happy to apply it.
> >
> >
> >
> > _**"While the runtime of MPNN + VN_G is the same as MPNN + VN, could there be problems with vanishing gradients since the depth is essentially doubled?"**_
> >
> > This shouldn’t be an issue as we are just performing a learnable global average between the layers and GNNs typically have a small number of layers. This can be seen by the number of layers (maximum 7) used in our hyperparameter tables (Table 4 and 5). We also find that this is not an issue in the experiments given that we see an improved performance using our approach.
> >
> >
> > Thanks again for your review and the comments which have helped improve the manuscript. We hope that we have sufficiently addressed your points and concerns. Please let us know if this remains unclear or if you consider any other discussion points to be open.

---

> > > ### Author Response · Authors · 2024-11-24
> > >
> > > Dear Reviewer aG7n,
> > >
> > > Given the otherwise relatively positive scores, we believe further discussion with you to be of great importance. Please let us know if our rebuttal has addressed your concerns or if you consider any of your concerns to remain open. Either way, we want to thank you very much for the time you have invested in reviewing our work thus far.

---

### Official Review · Reviewer_rwJ9 · 2024-11-02

**Soundness:** 3
**Presentation:** 3
**Contribution:** 3
**Rating:** 6
**Confidence:** 3

**Summary:**

(1) This paper studies the impact of Virtual Nodes (VNs) on oversquashing, showing that their improvements in network mixing can be bounded by the graph's spectrum.
(2) This paper identifies a gap between VNs and GTs in capturing heterogeneous node importance, leading to the proposal of MPNN + VNG, which effectively learns heterogeneous node relevance without extra cost.
(3) This paper validates its insights through experiments, demonstrating that MPNN + VNG consistently outperforms MPNN + VN, particularly in tasks where node heterogeneity is crucial.

**Strengths:**

1. Theoretical Contributions are good: (1) It offers the first comprehensive study of the impact of Virtual Nodes (VNs) on the oversquashing phenomenon, providing a foundational understanding of their role in enhancing network performance. (2) By employing sensitivity analysis of node features, the study identifies a significant gap between VNs and GTs regarding their ability to capture heterogeneous node importance, leading to deeper insights into their comparative strengths.
2. It introduces a novel MPNN + VNG method, that effectively utilizes graph structure to learn a heterogeneous measure of node relevance without incurring additional computational costs.
3. The paper validates its theoretical insights through extensive ablations and experiments, demonstrating that MPNN + VNG consistently outperforms the MPNN + VN, particularly in tasks where node heterogeneity is crucial.

**Weaknesses:**

1. ‘To assess if and how a VN helps to mitigate oversquashing, we need to determine whether the commute time of Gvn is smaller than the commute time of the original graph G.’ So which Theorem below can prove this?
2. Theorem 3.1 highlights how the impact of adding a VN can be determined in terms of the spectrum of the input graph. According to Theorem 3.1, how can it derive the claim of ‘adding a VN reduces the overall commute time’? Theoretically prove this is important.
3. ‘The result in Corollary 3.4 shows that for real-world graphs where adding a VN significantly reduces the commute time—i.e., (5) is negative’. Please further explain this claim.
4. ‘In our sensitivity analysis we will show MPNN + VNG to fall in between the fully homogeneous
MPNN + VN and the potentially fully heterogeneous GTs.’ Please give a brief explain for this claim.
5. ‘nodes that are more relevant to each of their neighbors, are more likely to contribute more to the global update, instead of weighting each node the same as for MPNN + VN.’ Which part of MPNN + VNG shows this property?
6. The heterogeneity of nodes is important, for what kind of tasks?

**Questions:**

See the weaknesses.

---

> ### Author Response · Authors · 2024-11-20
> **Response to Reviewer rwJ9**
>
> We thank the reviewer for their feedback and their response. Here, we try to respond to the weaknesses and questions point by point.
>
> _**"To assess if and how a VN helps to mitigate oversquashing, we need to determine whether the commute time of Gvn is smaller than ... the original graph. Which Theorem can prove this? According to Theorem 3.1, how can it derive the claim of adding a VN reduces the overall commute time?"**_
>
> The term to the left of the equality in Theorem 3.1 is the change in commute time between two nodes when we have a VN and when we don’t have a VN ($\tau_{\mathsf{vn}}(i,j) - \tau(i,j)$). Therefore, we can calculate whether the commute time of Gvn is smaller than the commute time of the original graph G using the term on the right. Previous works, for example by Di Giovanni et al. [1], have shown that oversquashing prevents the underlying model from exchanging information between nodes at large commute time. We can calculate the term on the right (which we show is negative on our datasets in Figure 1) to show that the commute time is reduced when adding a virtual node for certain datasets. We show in the manuscript that this means that adding a VN reduces the minimal number of layers required to learn functions with strong mixing between nodes (Theorem 3.3) which we demonstrate can lead to empirical performance gains.
>
> Further theoretical study on how this commute time is affected on particular subsets of graphs such as paths and trees, outside of the work we have done on real-world graphs, is challenging and remains a direction for future work.
>
> [1] Giovanni et al. How does over-squashing affect the power of gnns? Transactions on Machine Learning Research, 2024.
>
> _**"The result in Corollary 3.4 shows that for real-world graphs where adding a VN significantly reduces the commute time. Please further explain this claim."**_
>
> As discussed above, we can explicitly calculate the term on the right of the equality in Theorem 3.1 for real-world graphs using the graph spectrum. A negative term implies that the commute time of Gvn is smaller than the commute time of the original graph G. We calculate the graph spectra and show that for many real-world graphs this is negative (Figure 1) and thus using a VN reduces the commute time. Theorem 3.3 demonstrates the benefits of reducing the commute time by relating it to the expressivity of an MPNN in terms of learning graph functions between nodes with strong mixing. In Corollary 3.4, we combine Theorems 3.1 and 3.3 to show how a VN can reduce the number of layers required to learn such graph functions.
>
> _**"In our sensitivity analysis we will show MPNN + VNG to fall in between the fully homogeneous MPNN + VN and the potentially fully heterogeneous GTs.’ Please give a brief explain for this claim."**_
>
> In our sensitivity analysis we concern ourselves with the study of the Jacobian $\partial h_i^{(\ell+1)}/\partial h_k^{(\ell-1)}$ which allows us to determine whether the representation of a given central node $i$ depends on another node $k$ in the graph that has a shortest path distance greater than 2 to $i$. In Proposition 4.1 we show that the Jacobian MPNN+VN is independent of $k$. Then, in Proposition 4.2 we show that the Jacobian of MPNN+VN_G has a dependence on $k$ but it is independent of $i$. Finally, for the GT the Jacobian trivially depends on both $i$ and $k$, since all combinations of nodes are considered in the attention mechanism (this is now shown explicitly in Proposition F.1). Consequently, the MPNN+VN_G falls in between MPNN+VN and GT in this sensitivity analysis.
>
> _**"nodes that are more relevant to each of their neighbors, are more likely to contribute more to the global update, instead of weighting each node the same as for MPNN + VN.’ Which part of MPNN + VNG shows this property?"**_
>
> In the statement you quote we specifically refer to the ability of MPNN+VN_G to assign heterogeneous node importance. In particular, this statement is substantiated by the result derived in Proposition 4.2, where the two terms in which we sum over all nodes in the neighbourhood of node $k$, denoted by $N_k,$ are different for different nodes $k\in V$ resulting in heterogeneous node importance of nodes $k$ to the central node $i$.
>
> _**"The heterogeneity of nodes is important, for what kind of tasks?"**_
>
> We show in Table 1 that the importance of heterogeneous sensitivity is dataset dependent. We find that it is particularly relevant for CIFAR10 and causes a performance gap between MPNN+VN and GT. Heterogeneity allows us to understand **why** these two methods may differ and **provides us with a framework** to close this performance gap by introducing heterogeneity in the VN. However, it is difficult to know before running experiments which tasks/benchmarks would require more heterogeneity.
>
> Thanks again for your review and the comments which have helped improve the manuscript. We hope that we have sufficiently addressed your points and concerns.

---

> > ### Comment · Reviewer_rwJ9 · 2024-12-03
> >
> > The author's response addressed most of my concerns. Overall, the theoretical analysis is good, but it is not easy for readers to understand the meaning of these complicated formulas. Besides, the guidelines of when and how to apply these theories should be given. Therefore, I would like to maintain my rating.

---

### Official Review · Reviewer_cijV · 2024-11-03

**Soundness:** 3
**Presentation:** 3
**Contribution:** 3
**Rating:** 6
**Confidence:** 3

**Summary:**

The goal of this paper is to exploit the use of a virtual node to improve the performance of graph neural networks. The focus is specifically on the problem of oversquashing brought forward by the use of message passing neural networks (MPNNs). The authors first analyze the performance of MPNNs when virtual nodes are used and, then, they propose a new approach that can help improve performance and address the oversquashing problem.

**Strengths:**

+ The problem of oversquashing has been less studied in the prior art, and this work makes an important contribution towards this area
+ The analysis of oversquashing and the use of virtual nodes is important. The result connecting improvements to the graph spectrum is meaningful and helps in improving designs of graph neural networks.
+ The evaluation of performance compared to graph transformers is meaningful and shows how one can close the gap in performance.

**Weaknesses:**

- The theoretical results, while relevant, seem to follow directly from prior works. It is not clear whether the contribution is significant theoretically, as opposed to being a merger of known prior results like those from Di Giovanni et al.
- The argument for using MPNNs with virtual nodes compared to graph transformers is rather weak. I believe graph transformers may in fact remain the architecture of choice. The authors' discussion on this issue is not very convincing.
- The cost of adding a virtual node is apparently assumed to be negligible or at least it is not discussed.
- The performance gains do not seem significant enough

**Questions:**

- What is the cost of adding a virtual node? Are there hidden computing or performance costs? Can you provide concrete run-time or memory usage results while comparing the cases with and without virtual nodes?
- How do you justify the significance of your final results given that the performance is very close to state of art? For example, can you discuss the practical implications of the performance improvements that you achieved.
- Can you better articulate why MPNNs with virtual nodes are more effective than graph transformers?
- What is the main theoretical contributions when it comes to your proofs? How does it differ from the papers from which you adapted the theory? Please provide a point-by-point comparison between your theoretical results and the most closely related prior works.

---

> ### Author Response · Authors · 2024-11-20
> **Response to Reviewer cijV**
>
> We thank the reviewer for their feedback and their positive response. Here, we respond to the weaknesses and questions point by point.
>
> _**“The theoretical results, while relevant, seem to follow directly from prior works. How does it differ from the papers from which you adapted the theory? Please provide a point-by-point comparison between your theoretical results and the most closely related prior works."**_
>
> We think that the theoretical results outlined in this paper are comprehensively different from prior works. In Di Giovanni et al. they characterize the amount of mixing induced by an MPNN between two nodes after $m$ layers. There has been no prior work on how this mixing is affected by the addition of a virtual node (VN) to the graph. To compare to their results, we have to augment the initial graph with a VN and apply spectral analysis to the graph. This implies that Theorem 3.1 and Corollary 3.4 and our results showing how a virtual node impacts the mixing and (hence) oversquashing is completely novel and does not follow directly from prior works.
> Additionally, Section 4 provides theoretical results comparing GTs and MPNN+VN through sensitivity analysis which does not use results from prior works.
>
> _**“The argument for using MPNNs with virtual nodes compared to graph transformers is rather weak. I believe graph transformers may in fact remain the architecture of choice. Can you better articulate why MPNNs with virtual nodes are more effective than graph transformers?"**_
>
> This is a very interesting point. We would like to stress that we are not arguing in our paper for using MPNN+VN over Graph Transformers (GTs). We highlight a gap between the two approaches in terms of sensitivity and improve the standard MPNN+VN implementation to be closer to GTs in terms of our sensitivity analysis. In fact, GTs still allow for more heterogeneous sensitivity compared to our MPNN+VN_G as mentioned in Line 350. However, it is worth noting that our approach is more computationally efficient than GTs which can make it more suitable for certain applications. As discussed below, the complexity of a GT is quadratic $O(n^2)$ while that of an MPNN+VN is $O(|E| + n)$, where $|E|$ is the number of edges and is $<< n^2$ for real-world sparse graphs. In Appendix G.2, we have added the average number of nodes and edges in each of the datasets and also further clarified that $n$ is of the same order of magnitude as $|E|$ in these cases, hence $O(|E| + n)$ is much less than $O(n^2)$.
>
> We also find that MPNN+VN matches and sometimes even improves performance over GTs which may be overparameterized for the task. We even highlight why and when the performance will be similar (due to not requiring heterogeneous sensitivity for the task) in Table 1. In general, we believe that practitioners will consider a range of architectures that differ in terms of performance and complexity and MPNN+VN_G is shown to provide a good balance of these factors on the tasks studied in our manuscript.
>
> _**“The cost of adding a virtual node is apparently assumed to be negligible or at least it is not discussed. What is the cost of adding a virtual node? Are there hidden computing or performance costs? Can you provide concrete run-time or memory usage results while comparing the cases with and without virtual nodes?"**_
>
> This is a very good point. Given a graph with $n$ nodes and $|E|$ edges, GTs have quadratic complexity $O(n^2)$ and MPNNs have complexity $O(|E|)$. MPNN+VN and MPNN+VN_G have complexity $O(|E| + n)$, which is a significant improvement over quadratic complexity if the graph is sparse (as is the case for the benchmarks used in the paper and the majority of real-world graphs). We have added the complexity of the VN in line #282 of the revision and also a discussion of this complexity for these datasets in Appendix G.2. Additionally, we have run the training time per epoch and inference time for the Peptides dataset and compared this to GPS which is significantly slower. For Peptides-Func we have that
>
> | Method | Train (s/epoch) | Test (s) |
> |---|---|---|
> | GPS | 8.39 |0.611 |
> | GatedGCN+PE+VN_G | 4.01 | 0.321 |

---

> ### Author Response · Authors · 2024-11-20
> **Response to Reviewer cijV, pt 2**
>
> _**"The performance gains do not seem significant enough. How do you justify the significance of your final results given that the performance is very close to state of art? For example, can you discuss the practical implications of the performance improvements that you achieved."**_
>
> We politely disagree that the performance gains are not significant. Firstly, we significantly improve over a standard MPNN+VN, with the same computational cost, on **all** tasks. MPNN+VN is a popular and practical baseline often used in the community. Additionally, we get the top performing results on 4/8 tasks and the second best performing on 3/8 tasks. Given that we are comparing to some of the SOTA approaches, including GTs which are more computationally expensive, we feel that these results are certainly significant and interesting to the community.
>
> Thanks again for your review and the comments which have helped improve the manuscript. We hope that we have sufficiently addressed your points and concerns. Please let us know if this remains unclear or if you consider any discussion points to be open.

---

> > ### Comment · Reviewer_cijV · 2024-12-02
> >
> > I thank the authors for their response. This clarified my main issues. That said, I believe the work addresses a rather niche area and the results are limited. Thus, I maintain my original score.

---

### Official Review · Reviewer_JV8P · 2024-11-06

**Soundness:** 3
**Presentation:** 3
**Contribution:** 4
**Rating:** 8
**Confidence:** 3

**Summary:**

The paper contains a theoretical examination of VNs and their ability to counteract the oversquashing frequently present in MPNNs and a juxtaposition with GT methods, leading to a justification for an intermediate heterogeneous VN approach, VN_\textit{G}.
A novel analysis shows that VNs impact on the commute time between nodes of a graph, as a function of the graph spectrum, can reduce oversquashing. Empirically, VNs are shown to reduce commute time compared to MPNNs on real graphs, though is possible for it to increase. Commute time in turn affects mixing a model can induce after \textit{m} layers with mixing being the number of interactions between nodes needed to solve a task. When VNs reduce commute time, it is shown that strong mixing increases, reducing the size of \textit{m} and reducing oversquashing.

However, GTs can arbitrarily change commute time and further exceed the previous improvement, though they do incur significant overhead when compared to VNs. A novel sensitivity analysis of GTs and VNs follows which examines the Jacobian of both methods, showing that GTs allow for global heterogeneity via attention as each node depends on non-uniform functions of the values of all nodes at the previous layer. Contrary to this, VNs are independent for nodes further than 2 hops away and thus global information is applied homogeneously. It is worth noting that a full examination of the impact of GTs on commute time is left as a future research direction.
This leads to the proposal of VN_\textit{G} to bridge the gap between VNs and GTs. This takes the form of sequential updates. First a local update on nodes, then VN update leveraging the previous nodes, and a final node update as a function of both local and VN updates for each node. An examination of the Jacobian shows that the sensitivity of each node to non-neighbors becomes a function of the topology while appearing to maintain local bias.

The need for heterogeneity in VN_\textit{G} is empirically shown by examining the similarity of the attention matrix of a trained GT model on the nodes of several graphs. An increase in similarity implies reduced heterogeneity and therefore a reduced gap between the performance of GT and the (less) heterogeneous methods. The performance of models during experiments matches this observation.
In line with the discussion of the paper, experiments to compare MPNN + VN_\textit{G} with other baselines for graph-level tasks demonstrates improved performance over MPNN + VN on tasks requiring heterogeneity and matching performance of GT methods, which has some implication for the impact of long-range dependencies for tasks.

The appendix of the piece also includes an examination of MPNN + VN with PairNorm and demonstrates that, for the linear case, the benefits of MPNN + VN are not due to its ability to produce the results of PairNorm to reduce oversmoothing. In fact, when MPNN + VN replicates PairNorm, its expressivity is reduced.

**Strengths:**

The piece does an excellent job of addressing a shortcoming in the literature on when and why Virtual Node are helpful from a theoretical perspective. It provided an extensive juxtaposition with a competing method and identified a key component of the performance gap between the two in specific settings (graph-level tasks). An incremental approach that has begun to appear in other works is further discussed and the piece contains the first theoretical analysis to justify its use.
Additionally, with minimal additional work an appendix of the paper proving the benefit of VNs does not come from their ability to replicate PairNorm for the linear case, which is rather a hindrance.

**Weaknesses:**

I do not believe that there are any significant weaknesses to this paper and find it to be acceptable. Minor changes may improve the piece, but also to an undue amount of additional work and conflict with length requirements.

MPNN + VN can beat GTs as shown in Tables 1 and 3 with the observation that long-range dependencies may not play as large of a role for these tasks. This leads me to wonder about the role of the inductive bias from the locality of MPNN’s, though this is addressed Rampášek et al. which instead leads be to the impact of VN_\textit{G} on mixing.

I do not have a better alternative, but the empirical analysis to answer Q2 of Section 5.1 oddly seems self-justifying to me. In addition, GatedGCN+PE is shown in Table 2. to have superior performance to all the methods of Table 1. For Pept-Func while remaining competitive for Pept-Struct.

It seems that Appendix C concludes the proof of Theorem B.1 but only proves that there are filters which can be learned by MPNN but not MPNN-VN. The remainder in this seems mostly to be contained in Theorem C.1.

**Questions:**

Beginning on line #1430 are what appear to be notes used to write Appendix C that were not removed.

Line #373 refers to Theorem 3.4 and should instead be Corollary 3.4.

It would be nice to have the Jacobian of GT at least somewhere in the appendix.

In Table 1 what is the PE in GatedGCN+PE+VN? Also, in the text of 5.1 it is only referred to as GatedGCN+VN

I was not able to discern what the arrows in the heading of each of the tables indicated.

Appendix J seems unnecessary as Fig. 1 would be more useful in the main text, if it wasn’t so small.

---

> ### Author Response · Authors · 2024-11-20
> **Response to Reviewer JV8P**
>
> Thank you very much for your attentive review. We very much appreciate your positive feedback.
>
> _**“This leads me to wonder about the role of the inductive bias from the locality of MPNN’s, though this is addressed Rampášek et al. which instead leads be to the impact of VN_G on mixing.”**_
>
> Interestingly, recent work has demonstrated the effectiveness of the locality bias of MPNNs on these tasks [1]. As we demonstrate in our Section 3, MPNN+VN improves the mixing abilities of the network whilst maintaining this locality bias and hence can improve performance further (Table 3). We also maintain this locality bias whilst introducing heterogeneous sensitivity to the network with MPNN+VN_G. Again, we show the empirical performance improvements by doing this in Tables 2 and 3. So, we agree that this is an interesting consideration, which provides a nice perspective on our contributions.
>
>
> _**“I do not have a better alternative, but the empirical analysis to answer Q2 of Section 5.1 oddly seems self-justifying to me. In addition, GatedGCN+PE is shown in Table 2. to have superior performance to all the methods of Table 1. For Pept-Func while remaining competitive for Pept-Struct.”**_
>
> In Q2, we are looking to see if heterogeneity in the global update is more impactful on some datasets than others. To do this, we remove the heterogeneity in the transformer of GPS (“GPS + projection” in Table 1). Interestingly, we find that on some datasets (where the std of the attention matrix is low) this does not affect performance, thus providing an answer to Q2.
> You are right that GatedGCN+PE has very good performance on these tasks which was also demonstrated and argued for in [1]. Introducing our VN with heterogeneous sensitivity outperforms this on all tasks though, suggesting that improving the mixing ability of the network and allowing node heterogeneity in the VN can be helpful.
>
>
> _**“It seems that Appendix C concludes the proof of Theorem B.1 but only proves that there are filters which can be learned by MPNN but not MPNN-VN. The remainder in this seems mostly to be contained in Theorem C.1."**_
>
> Please allow us to clarify the structure of the concerned proof of Theorem B.1 in Appendix C. In Lines 787-89 we show that any filter learned by an MPNN can also be learned by an MPNN+VN. Then, in Lines 810-16 we show that there exists filters that can be learned by MPNN+VN but not by MPNN. Finally in Lines 783-816 we show that there are filters that can be learned by MPNN but not MPNN-VN, as you saw. It seems to us that the proof is therefore complete and covers all statements made in the theorem.
>
> _**“notes used to write Appendix C that were not removed. Line #373 refers to Theorem 3.4."**_
>
> Thank you for spotting this! We have removed these notes. Additionally, we have changed line #373 to mention the Corollary in the revision.
>
> _**“It would be nice to have the Jacobian of GT"**_
>
> We agree that this Jacobian completes our argument nicely and have therefore now included it in Lines 1112-47 in Appendix F Proposition F.1.
>
>
> _**“In Table 1 what is the PE in GatedGCN+PE+VN?"**_
>
> For the GPS model and its projection, we used the final hyperparameters as described in the original work [2]. In their work, they optimize the positional encoding (PE) from the choice of [None, LapPE, RWSE]. For the different datasets in Table 1, the optimal PE is always found to be the Laplacian positional encoding (LapPE).
>
> For GatedGCN+PE+VN, we follow the same process as GPS [2] where the PE is chosen from [None, LapPE, RWSE]. This hyperparameter optimization is done as described in Appendix G.4. To help with the clarity of these experiments, we have added a discussion specifically about the Table 1 hyperparameters in Appendix G.4.
>
>
> _**“I was not able to discern what the arrows in the tables indicated"**_
>
> The arrows indicate the direction of the scores which we are trying to optimise for. For example, if we aim to reduce Mean Absolute Error, we would have a downarrow and if we want to maximise Accuracy, we would have an uparrow. We have added a description highlighting what the arrows mean in the caption of Table 1.
>
>
> _**“Appendix J seems unnecessary."**_
>
> We hope that the text and our model equations have clearly explained the model architecture. We added Appendix J with the model figure in order to complete and provide additional material to complement the text. We don’t feel that the figure would add sufficient value in the main text so have left it as an appendix item.
>
> We hope that we have managed to respond to your questions and we again thank the reviewer for the effort which has been put in.
>
> [1] Tönshoff et al. Where Did the Gap Go? Reassessing the Long-Range Graph Benchmark. TMLR 2024.
>
> [2] Rampášek et al. Recipe for a General, Powerful, Scalable Graph Transformer. NeurIPS 2022.

---

### Author Response · Authors · 2024-11-20
**General response**

We are very grateful to all reviewers for their comprehensive responses which, we gladly note, have highlighted several positive traits of our presented work.

The work is said to have made an “important contribution” [cijV] which has helped in “addressing a shortcoming in the literature” [JV8P] and provided the “first” [rwJ9, JV8P] “comprehensive” [rwJ9, kkhT] study on the impact of a VN on oversquashing. We have also “identified a key component” [JV8P] of the performance gap between MPNN+VN and GTs. The validation of the theoretical findings are also said to be “extensive” [rwJ9] and “meaningful” [cijV].

The reviewers have also kindly provided interesting points of discussion and raised relevant questions which we address specifically below. A new revision has been posted and includes improvements based on reviewers’ comments.

---

### Meta-Review · Area_Chair_npwY · 2024-12-20

**Metareview:**

This paper investigates how adding a virtual node affects MPNN regarding oversquashing and sensitivity to distant nodes. It provides a theoretical framework that relates commute times in graphs to the presence of VNs, offering precise spectral conditions under which VNs help reduce oversquashing. Authors propose also a variant of the VN mechanism that can exploit heterogeneous node importance without incurring extra computational cost, narrowing the gap with more complex graph transformer architectures.

Strengths:  The work gives an interesting interpretation of of virtual nodes in MPNNs, not only by confirming that they can mitigate oversquashing but also by characterizing when and why this occurs through spectral analysis. This contributes new insights into which kinds of graph topologies benefit most from VNs. The proposed VN variant outperforms standard baselines without increased complexity, showing that the insights from theory can translate into (potential) practical improvements. The paper is generally well-structured, with good evidence supporting its claims.

Weaknesses: Some reviewers considered the theoretical contributions more incremental than groundbreaking, noting that the results build closely on established concepts. I agree with them, but still feel that it a nice interpretation of hese results. Others felt that certain formulas and theoretical aspects were complex, offering limited practical guidance. I believe this is expected for this kind of work. Additionally, while the authors compared their approach to GTs, some found the comparison lacking in depth. The reported empirical gains, though consistent, are not dramatic, leading some to question the overall impact. This is true, but again, I don't think the main point of the paper is to provide SOTA results, as the title indicate: "Understanding Virtual Nodes". In this sense, I think the paper is well aligned with its objectives.

Overall, I believe that the paper makes a valuable contribution to the understanding of virtual nodes in MPNNs, and I recommend acceptance despite those concerns.

**Additional Comments On Reviewer Discussion:**

The communication during the rebuttal was quite limited. As addressed in my metareview, I believe most of the issues raised by the referees were mitigated during the rebuttal.

---

### Decision · Program_Chairs · 2025-01-22

Accept (Poster)